# High frequency measurements explain quantity and quality of dissolved organic carbon mobilization in a headwater catchment

Benedikt J. Werner[1], Andreas Musolff[1], Oliver J. Lechtenfeld[2], Gerrit H. de Rooij[3], Marieke R. Oosterwoud[1], Jan H. Fleckenstein[1]

[1]Department Hydrogeology, Helmholtz Centre for Environmental Research - UFZ, Leipzig, 04318, Germany
[2]Department Analytical Chemistry, Research group BioGeoOmics, Helmholtz Centre for Environmental Research - UFZ, Leipzig, 04318, Germany
[3]Department Soil System Sciences, Helmholtz Centre for Environmental Research - UFZ, Halle, 06120, Germany

*Correspondence to*: Benedikt J. Werner (benedikt.werner@ufz.de)

**Abstract.** Increasing dissolved organic carbon (DOC) concentrations and exports from headwater catchments impact the quality of downstream waters and pose challenges to water supply. The importance of riparian zones for DOC export from catchments in humid, temperate climates has generally been acknowledged, but the hydrological controls and biogeochemical factors that govern mobilization of DOC from riparian zones remain elusive. A high-frequency dataset (15 minutes resolution for over one year) from a headwater catchment in the Harz Mountains (Germany) was analyzed for dominant patterns in DOC concentration ($C_{DOC}$) and optical DOC quality parameters $SUVA_{254}$ and $S_{275-295}$ (spectral slope between 275 nm and 295 nm) on event and seasonal scale. Quality parameters and $C_{DOC}$ systematically changed with increasing fractions of high-frequency quick flow ($Q_{hf}$) and antecedent hydroclimatic conditions, defined by the following metrics: Aridity Index ($AI_{60}$) of the preceding 60 days, and the quotient of mean temperature ($T_{30}$) and mean discharge ($Q_{30}$) of the preceding 30 days which we refer to as discharge-normalized temperature ($DNT_{30}$). Selected statistical multiple linear regression models for the complete time series ($R^2$= 0.72, 0.64 and 0.65 for $C_{DOC}$, $SUVA_{254}$ and $S_{275-295}$, resp.) captured DOC dynamics based on event ($Q_{hf}$ and baseflow) and seasonal-scale predictors ($AI_{60}$, $DNT_{30}$). The relative importance of seasonal-scale predictors allowed for the separation of three hydroclimatic states (warm & dry, cold & wet and intermediate). The specific DOC quality for each state indicates a shift in the activated source zones and highlights the importance of antecedent conditions and its impact on DOC accumulation and mobilization in the riparian zone. The warm & dry state results in high DOC concentrations during events and low concentrations between events and thus can be seen as mobilization limited, whereas the cold & wet state results in low concentration between and during events due to limited DOC accumulation in the riparian zone. The study demonstrates the considerable value of continuous high-frequency measurements of DOC quality and quantity and its (hydroclimatic) key controlling variables in quantitatively unraveling DOC mobilization in the riparian zone. These variables can be linked to DOC source activation by discharge events and the more seasonal control of DOC production in riparian soils.

# 1 Introduction

Dissolved organic carbon (DOC) in streams is a significant part of the global carbon cycle (Battin et al., 2009) and plays a vital role as a nutrient for aquatic ecosystems. Riverine exports of DOC from catchments can impair downstream aquatic ecology and water quality (Hruska et al. 2019) with potential implications for the treatment of drinking water from surface water reservoirs (Alarcon-Herrera et al., 1994). The pivotal role of DOC for surface water quality and ecology is not only related to the concentration ($C_{DOC}$) in the water, but also to the specific chemical composition of DOC, referred to here as DOC quality. For example, DOC quality defines the thermodynamically available energy (Stewart and Wetzel, 1981), which in turn affects the growth of microorganisms (Ågren et al., 2008). Consequently, changes in DOC quality could change the patterns of aquatic microbial metabolism resulting in altered ecosystem functioning (Berggren and del Giorgio, 2015). For managing water quality and aquatic ecology in surface waters it is therefore not only important to understand the drivers and controls of DOC concentration, but also of the associated DOC quality. This study takes a step in this direction.

DOC concentrations in streams were found to be highly variable in time with strong controls being discharge (Zarnetske et al. 2018), climatic conditions (Winterdahl et al., 2016), or at longer timescales the prevailing biogeochemical regime (Musolff et al., 2017). DOC concentration variability is also closely linked to distinct DOC source zones in catchments and their hydrologic connectivity to the stream network (Broder et al. 2015, Birkel et al. 2017). In temperate humid climates most of the riverine DOC export is typically derived from terrestrial sources at or near the terrestrial-aquatic interface (Laudon et al., 2012; Ledesma et al., 2018; Musolff et al., 2018; Zarnetske et al., 2018). More specifically, the riparian zone is seen as a dominant source zone for DOC, defining potential DOC export loads and their temporal patterns (Ledesma et al., 2015; Musolff et al., 2018). In this zone, DOC export is strongly controlled by lateral hydrologic transport through shallow organic-rich soil layers thus connecting different patches of differently processed DOC pools to the stream. The capacity of the riparian zone to drain and produce discharge and thus export DOC generally increases with the rise of the groundwater table during events. This causes a non-linear increase in the lateral transmissivity of the riparian soil profile and the resulting subsurface flux to the stream, which has been called the transmissivity feedback mechanism (Bishop et al., 2004; Rodhe, 1989). However, also distinct preferential flow paths in the subsurface (Hrachowitz et al., 2016) and at the surface (Frei et al., 2010) can play a considerable role. The associated DOC export to the streams was found to be mostly transport limited (Zarnetske et al., 2018) with storm events generally generating most of the overall loads exported from catchments (Buffam et al., 2001; Hope et al., 1994). Daily precipitation and amount of discharge were found to be event-scale drivers (Bishop et al., 1990) defining magnitude and timing of DOC export. Strohmeier et al. (2013) therefore pointed at the importance of temporally resolved concentration measurements for accurate load estimates.

Besides discharge and transport capacity the biogeochemical regime in the riparian soils, which controls the build-up, size and quality of the exportable DOC pool was identified as an additional important control for DOC export from catchments (Winterdahl et al., 2016). This build-up of exportable DOC pools in turn is strongly related to the hydroclimatic conditions like temperature and soil moisture content prior to an event (Birkel et al., 2017; Broder et al., 2017; Christ and David, 1996;

Garcia-Pausas et al., 2008; Preston et al., 2011), which to some degree also define the potential for hydrological connectivity and transport during the event (Birkel et al., 2017; Köhler et al., 2009; Shang et al., 2018). On a seasonal scale (roughly $1-3$ months) hydroclimatic variables control intra-annual variability of DOC concentration and quality (Ågren et al., 2007; Hope et al., 1994; Köhler et al., 2009) and are hence considered as important drivers of seasonal DOC export dynamics (Ågren et

al., 2007; Birkel et al., 2014; Köhler et al., 2009; Seibert et al., 2009). In summary, DOC concentration and quality are jointly controlled by the hydrologic conditions during events (defining the timing and magnitude of DOC export) and the antecedent hydroclimatic conditions (defining size and quality of exportable DOC pools in the soil), resulting in a highly dynamic system with processes interacting at time-scales ranging from the event-scale of hours to days to timescales of seasons. Characterizing and quantifying such a dynamic system requires measurements of DOC concentration and quality at

a sufficient temporal resolution. Yet, most studies to date have only focused on temporally aggregated data (Köhler et al., 2008) and the seasonal to annual time scale with little or no consideration of the strong interaction with event-scale variability of DOC quantity and quality (Bishop et al., 1990; Strohmeier et al., 2013).

Recent years have seen significant advances in sensing technologies for high-frequency in situ concentration measurements (Rode et al., 2016; Strohmeier et al., 2013), facilitating the assessment of the highly dynamic DOC delivery to streams

(Tunaley et al., 2016). Differences in DOC quality observed during varying runoff conditions have been used to characterize source zone activation in smaller watersheds (Hood et al., 2006; Sanderman et al., 2009). Hence, the combination of high frequency $C_{DOC}$ measurements with additional spectral and analytical methods to characterize DOC quality (Herzsprung et al., 2012; Raeke et al., 2017; Roth et al., 2013) at temporal resolutions capable of capturing the dynamics within hydrologic events provides an opportunity to significantly improve our mechanistic understanding of DOC mobilization, transport, and

ultimately export from catchments (Berggren and del Giorgio, 2015; Creed et al., 2015; Köhler et al., 2009; Strohmeier et al., 2013). Broder et al. (2017) jointly evaluated DOC concentration and quality dynamics, but they were limited to hourly event data and bi-weekly data between events. Here we see great potential in the systematic analysis of high frequency data for improving our understanding of the delicate interplay between hydrologic (mobilization and transport) and biogeochemical controls (build-up of exportable DOC pools) from the event to seasonal scales that ultimately control DOC export from

catchments. This could also stimulate improvements in the formulation of models for DOC export to streams, which are often constrained in terms of transferability across spatiotemporal scales because of a mismatch between the scales of observations and that of the underlying processes (Zarnetske et al., 2018).

We hypothesize that seasonal- and event-scale DOC quantity and quality dynamics in headwater streams are dominantly controlled by the dynamic interplay between event-scale hydrologic mobilization and transport (delivery to the stream) and

inter-event and seasonal biogeochemical processing (exportable DOC pools) in the riparian zone. Furthermore we hypothesize that continuous high-frequency measurements of $C_{DOC}$ and spectral properties can be utilized to identify and quantify the key controls of DOC quantity and quality dynamics. The objectives of this study are 1) to use high-frequency in-stream observations of DOC quantity and quality during different seasons to elucidate the effects of hydroclimatical factors (which include frequency and intensity of rainfall and snowmelt events) on mobilization and export of DOC; and 2)

to establish a set of key controlling variables that captures important hydrologic, hydroclimatic and biogeochemical characteristics of the system to allow a quantitative assessment of stream DOC quantity and quality during different times of the year.

To this end, a high-frequency dataset on $C_{DOC}$ and DOC quality from a first-order stream in Central Germany was evaluated in terms of key controlling variables such as discharge, temperature and antecedent wetness conditions. The dominant drivers of seasonal- and event-scale variability of $C_{DOC}$ and quality were extracted and assessed (a) by a correlation analysis of intra-annual variations (seasonal scale ≥ 1 month), and (b) by an analysis of the individual discharge events throughout the year (event scale, hours - days), respectively. In a final step (c), these drivers were interpreted mechanistically based on a multiple linear regression analysis covering the entire study period. The identified parameters are discussed with respect to underlying processes and synthesized in a conceptual model of DOC export.

## 2 Materials and Methods

### 2.1 Study site

Measurements were conducted in a headwater catchment of the Rappbode stream (51°39'22.61"N 10°41'53.98"E, Fig. 1) located in the Harz Mountains, Central Germany. The Rappbode stream flows into a large drinking water reservoir. Downstream of the reservoir it flows into the river Bode, and eventually discharges (via the rivers Saale and Elbe) into the North Sea. The investigated part of the catchment has an area of 2.58 km² and a drainage density of 2.91 km km$^{-2}$. The catchment is mainly forested with spruce and pine trees (77%), the remaining area is covered with grass (11%) and other vegetation (12%). Elevation ranges from 540 to 620 m above sea level; the mean topographic slope is 3.9°. The 90[th] percentile of the topographic wetness index as a measure for the extent of riparian wetlands in the catchment (Musolff et al., 2018) is 8.53 (median 6.77). The geology at this site consists mainly of graywacke, clay schist and diabase (Wollschläger et al., 2016). Soils in the vicinity of the Rappbode spring are dominated by peat. Overall, one quarter of the catchment is characterized by groundwater-influenced humic gleysols and stagnic gleysols, which are mainly found in the riparian zones. Riparian soils were mapped next to the Rappbode stream, 2 km downstream of the spring (Fig. 1). At this site, topsoil layer (A horizon) thickness in a transect was 17.7 cm +/- 2.4 cm on average (n = 27) up to 25 m off the stream. The study site has a temperate climate (Kottek et al., 2006), with a long-term mean temperature of 6.0 °C and mean annual precipitation of 831 mm (Stiege weather station 12 km away from the study site, data provided by the German Weather Service DWD).

### 2.2 Data basis

An overview of all variables utilized for site description and regression modeling as well as descriptive statistics of these variables are given in Table 1.

### 2.2.1 Monitoring of response variables: DOC concentration and quality

We used in situ absorption spectroscopy to estimate dissolved organic matter quantity and quality. For simplification and because carbon is the main focus of the paper, dissolved organic matter quality will be addressed as DOC quality in the following. DOC quality can be characterized by specific metrics based on the light absorbing properties of dissolved organic compounds: $SUVA_{254}$ was calculated by normalizing the spectral absorption coefficient at 254 nm ($SAC_{254}$) for the according $C_{DOC}$ values. $SUVA_{254}$ correlates well with aromaticity of DOC and therefore can be used as an indicator of the general chemical composition and reactivity of organic carbon (Weishaar et al., 2003). To refine the understanding of DOC composition, the spectral slope between 275 and 295 nm, denoted $S_{275-295}$ was estimated from the adsorption spectra and calculated as described in Helms et al. (2008): A linear regression model was fitted for each time step to the logarithms of the absorption coefficients between 275 and 295 nm to derive the slope $S_{275-295}$. $S_{275-295}$ can help to distinguish between autochthonous and allochthonous DOC, molecular weights and processing (photobleaching and microbial degradation change aromaticity) (Helms et al., 2008). The general patterns of such DOC quality metrics can be used to infer information on origin and properties of DOC and thus to characterize source zones of DOC in riparian zones (Eran et al., 2006; Hutchins et al., 2017; Sanderman et al., 2009). An UV-Vis probe (Spectrolyzer, s::can Messtechnik GmbH, Austria) was installed in the stream (Fig. 1) from April 2013 until October 2014 to measure light absorption spectra from 220 nm to 720 nm in 2.5 nm steps every 15 min. There is a data gap from 11 December 2013 until 14 January 2014 due to general maintenance and recalibration of the UV-Vis probe in the laboratory. Other gaps from 18 November 2013 until 27 November 2013 and from 01 September 2014 until 17 September 2014 were due to a probe failure; accordingly values were excluded a priori. Overall, the UV-Vis dataset comprises 42,427 measurements. For a description of fouling correction, onsite probe maintenance and sampling procedure refer to S1 in the supplements.

After fouling correction, UV-Vis measurements were used to derive $C_{DOC}$, $SUVA_{254}$ and $S_{275-295}$. For validation and calibration of $C_{DOC}$ and $SUVA_{254}$, 28 grab samples were used that have been taken biweekly from the stream to measure the specific absorption coefficient at 254 nm ($SAC_{254}$ (UVT P200, Real Tech Inc., Canada). Subsequently, $C_{DOC}$ was measured in the laboratory by thermo-catalytic oxidation at 900°C with NDIR detection (DIMATOC® 2000, Dimatec Analysentechnik GmbH, Germany). A continuous time series of $C_{DOC}$ from the UV-Vis spectra was created using partial least squares regression (PLSR) to the 28 concentration values via the R package pls (Mevik and Wehrens, 2007). The PLSR proved to robustly work with a large number of predicting variables and strong collinearities (Musolff et al., 2015; Vaughan et al., 2017). The procedure generally followed the method described in Etheridge et al. (2014) using all turbidity-compensated spectra within a single regression model, chosen by 10-fold cross validation of the training data set. Through this method, $C_{DOC}$ was defined by a local combination of several wavelengths that proved to yield better results than the predefined global settings provided by the probe (Vaughan et al., 2017).

$SUVA_{254}$ was calculated by dividing the spectral absorption coefficient at 254 nm ($SAC_{254}$) by the PLSR-derived $C_{DOC}$ values. The resulting $SUVA_{254}$ values were then validated (but not calibrated) by the 28 $SUVA_{254}$ values derived from the manual

$SAC_{254}$ measurements in the field and the associated lab $C_{DOC}$ measurements (see 3.1). As second quality metric $S_{275-295}$ was estimated from the fouling-corrected adsorption spectra as described above and in Helms et al. (2008). There are no laboratory values available to verify $S_{275-295}$ calculations, so calculated values were verified by comparison to the literature.

### 2.2.2 Predictor variables: Stream level and discharge, evapotranspiration and antecedent wetness condition

Discharge $Q_{tot}$ was calculated from a stage-discharge relationship, which was established based on the 15 min stage readings from a barometrically compensated pressure transducer (Solinst Levellogger, Canada) and biweekly manual discharge measurements using an electromagnetic flow meter (n = 42; MF pro, Ott, Germany).

Manually measured discharge maximum was 0.39 m³ s$^{-1}$ at a water level of 83.8 cm. Ungauged water levels above this value and the associated discharges were extrapolated from the stage-discharge relationship and found to be within a valid range

when comparing to modelled discharge from the mesoscale hydrological model mHM (Mueller et al., 2016; Samaniego et al., 2010). A hydrograph separation into event and baseflow components was applied following the method described by Gustard and Demuth (2009). Total discharge $Q_{tot}$ was partitioned into a high-frequency quick flow ($Q_{hf}$) component, active during events and a low frequency component representing base flow ($Q_b$). To derive the baseflow hydrograph, local flow minima of non-overlapping five-day periods were selected and linearly connected to each other using the lfstat package

(Koffler et al., 2016) in R (R-Core-Team, 2017). If the baseflow hydrograph exceeded the actual flow, it was constrained to equal the observed hydrograph of $Q_{tot}$. Consequently, subtracting the baseflow hydrograph ($Q_b$) from the total hydrograph of $Q_{tot}$ yields the hydrograph of $Q_{hf}$, which has positive values during events ($Q_{tot} > Q_b$) and zero values during non-event periods (when $Q_{tot} = Q_b$). All consecutive positive values between two non-event periods (zero values) were considered as one event and extracted from the complete dataset for further processing.

To characterize ambient weather conditions, a weather station (WS-GP1, Delta-T, United Kingdom) placed about 250 m northwest of the UV-VIS probe provided data on air temperature ($T$), air humidity, wind direction and speed, solar radiation, and rainfall ($P$) at a 30 min interval. Measurements of the weather station started at 21 May 2013 until 26 November 2014. Measurements were at an hourly interval for the first five days, until 26 June 2013.

Potential evapotranspiration ($ET_P$) was calculated on an hourly basis from the weather data after Penman-Monteith (Allen et

al., 1998). The antecedent aridity index ($AI_t$) gives an estimate of the water balance in the last $t$ days and equals the aridity index for longer time periods given by Barrow (1992). Accordingly, $AI_{60}$ was derived for the measurement period by dividing the cumulative sum of precipitation over the last 60 days ($P_{60}$) by the cumulative sum of $ET_P$ of the last 60 days ($ET_{P60}$). As a consequence, time series of lumped variables start $t$ days after the actual begin of the field observations.

The discharge-normalized temperature of the preceding 30 days ($DNT_{30}$) was calculated by dividing the mean air

temperature of the preceding 30 days by the mean discharge of the preceding 30 days. $DNT_{30}$ gives an estimate of the ratio between temperature (that controls soil DOC production, e.g. Christ and David (1996)) and discharge (that controls DOC export, e.g. Hope et al. (1994)) in the last 30 days and therefore can potentially be related to the state of DOC storage in top

soils. We chose $AI_{60}$ and $DNT_{30}$ as these variables turned out to work best in terms of variance inflation and interaction for the statistical modeling.

In order to obtain an analogous dataset, time series of all variables were constrained by excluding such observations that fell into the data gaps of the UV-Vis probe (cf. 2.2.1).

## 5  2.3 Statistical analysis

Evaluation of the variable's predictive power was done for the entire dataset as well as for separated discharge events. Descriptive statistical tools were applied using the software R (R-Core-Team, 2017). Spearman's rank correlation ($r_s$) was used to look for significant relations of $C_{DOC}$ and DOC quality with potential controlling variables, since concentration, discharge and solute loads in river systems usually have lognormal probability distributions while C-Q relationships can be
described by power law functions (Jawitz and Mitchell, 2011; Köhler et al., 2009; Rodríguez-Iturbe et al., 1992; Seibert et al., 2009).

### 2.3.1 Event-scale analysis

Consequently, concentration-discharge (C-Q) relationships were characterized and quantified in log-log space for the event analysis. Since metrics of DOC quality are typically normally distributed (Guarch-Ribot and Butturini, 2016; Sanderman et
al., 2009), relationships between quality and $Q_{tot}$ were analyzed in semi-log space. According C-Q and quality-Q relationships for each runoff event (n = 38, extracted with the method explained in 2.2.2) were represented by combinations of multiple linear regression models with $Q_{tot}$, $Q_b$ and $Q_{hf}$ and their log transformations as predictors. As recommended by Marquardt (1970) and Menard (2001), multicollinearity of predictors was taken into account based on the variance inflation factor (VIF; R package car (Fox and Weisberg, 2011)):

$$VIF_i = \frac{1}{1-R_i^2} > 10 \tag{1}$$

where $VIF_i$ is the variance inflation factor for every predictor variable $i$ in the complete model, predicted by multiple linear regression from the remaining predictor variables of the complete model. $R_i^2$ is the corresponding coefficient of determination. Predictor variables were excluded from the model if Eq. (1) holds for predictor variable $i$.

The best overall combination of two variables for the prediction of events was chosen according to the best mean $R^2$ of all 38
single models. Hence, independent variable $\log(C_{DOC})$ is best predicted by a combination of both discharge components ($\log(Q_{hf})$ and $Q_b$) during single discharge events. Subsequently, the 38 triplets of intercepts and regression coefficients of these single models were extracted for further analysis. Note that the hysteresis loop size did not significantly bias regression coefficients obtained from this method (S2, Fig. S1).

### 2.3.2 Seasonal-scale analysis

To explain seasonal variations in the event analysis, the 38 regression coefficient triplets were correlated with seasonal-scale antecedent key controlling variables. Variables which showed strong correlations were added in different combinations to the existing event model as potential predictors for seasonal variations in addition to the event-scale variance. Here, models of the dependent variables ($C_{DOC}$, $SUVA_{254}$ and $S_{275-295}$) models always used the same predictor variables. The interaction between two predictor variables was generally used for modelling. This implies that the measured hydroclimatic variables influence each other and thus cause a non-additive effect on the dependent variable. Here, we write interaction terms as the product between the two interacting variables (variable1 × variable2). Again, predictors (variables and interaction terms) were tested for multicollinearity and excluded from the complete model if Eq. (1) holds for variable $i$.

Akaike's Information Criterion (AIC) and R² were used for model selection and validation. Five-fold cross-validation was applied to estimate the prediction error. Once the most valid model was selected, the predictive power of the chosen predictors for the different models of $C_{DOC}$ and DOC quality was tested. Partial models were built by stepwise dropping the least influencing predictors according to AIC and by comparing the subset of event-scale predictors with the subset of seasonal-scale predictors.

## 3. Results

### 3.1 Monitoring of DOC and hydroclimatic parameters

The basic statistics of UV-Vis-derived $C_{DOC}$ and DOC quality as well as hydroclimatic variables throughout the 1.5-year measurement period are given in Table 1.

The amount of precipitation during 2013 (665 mm) and 2014 (682 mm) was close to the long-term annual mean at the nearest weather station. Discharge shows event-type variability but followed in general the hydrological year, with lowest values in late summer and highest values in spring (Fig. 2a). Highest discharge was 1.98 m³ s$^{-1}$ during snowmelt on 27 April 2014. With a coefficient of variation (CV) much higher than 1, the discharge regime can be described as erratic (Botter et al., 2013), indicating the importance of the quick flow component for discharge in the Rappbode catchment. Consequently, the variability of $Q_{hf}$ mostly follows $Q_{tot}$, but without the seasonal baseflow trends. A total number of 38 discharge events have been separated by discharge partitioning, yielding an average frequency of 0.086 d$^{-1}$ (2.58 month$^{-1}$) at an average duration of 134 h per discharge event. A dry period occurred from 14 June 2013 to 23 July 2013, which resulted in a steady decline in discharge during that time (Fig. 2).

Air temperature exhibited strong seasonal patterns and was comparable to the seasonal mean at the nearest station. Daily sums of $ET_P$ peaked in summer whereas $ET_P$ in autumn and winter reached the minimum. The general pattern follows a typical seasonal sinusoidal shape (not shown).

The aridity index $AI_{60}$ (median = 1.43) indicates a general wet climate with higher precipitation than potential evapotranspiration. $AI_{60}$ peaked in winter whereas minimum values occurred in summer during the drought and in winter during the freezing period (Fig. 2b). Summer precipitation has only little impact on $AI_{60}$. With a CV of 0.74, $ET_{P60}$ generally has more influence on the variability of $AI_{60}$ than $P_{60}$ (CV = 0.53).

$DNT_{30}$ peaked in summer whereas minimum values occurred in winter (Fig. 2b). Generally, $Q_{30}$ (CV = 0.89) has more influence on the variability of $DNT_{30}$ than $T_{30}$ (CV = 0.53). Precipitation events in cold periods have only little impact on $DNT_{30}$ and peaks due to precipitation are barely detectable.

$C_{DOC}$ based on the PLS regression fits well to the DOC concentration measured in the lab ($R^2$ = 0.97, residual standard error: 0.68 mg L$^{-1}$) (Fig. 2c). The maximum deviation of PLS-based $C_{DOC}$ from lab-measured $C_{DOC}$ was 1.7 mg L$^{-1}$ on 24 July 2013. We argue that the PLSR predicts the average characteristic composition of DOC rather well but hardly accounts for changes in DOC quality and thus spectral properties due to extreme situations like droughts and floods which can strongly differ in DOC source area mobilization in comparison to average events (Vaughan et al., 2017). Accordingly, $C_{DOC}$ and hence calculated $SUVA_{254}$ values match the manual measurements to a lesser extend during such situations, leading to an overall $R^2$ of 0.5 for $SUVA_{254}$ values, but removing three measurements taken during longer dry periods (09 July 2013, 04 September 2013, 23 July 2014) increases overall $R^2$ to 0.73.

There are no laboratory values available to verify $S_{275-295}$ calculations, but calculated values are in the same magnitude as reported in the literature (Helms et al., 2008; Spencer et al., 2012).

$C_{DOC}$, $SUVA_{254}$ and $S_{275-295}$ exhibit pronounced event-type variability over the entire year (Fig. 2c - e). In winter months, DOC was low in concentration, but had a distinct quality signature with high $S_{275-295}$ and $SUVA_{254}$ values (Fig. 2c - e). Furthermore, only small fluctuations of concentration and quality were observed in winter. Summer months showed minimum $C_{DOC}$, $SUVA_{254}$ and $S_{275-295}$ values in both years during baseflow, but also the most distinct $C_{DOC}$ and quality variations during discharge events. Late summer and autumn $C_{DOC}$ were different between 2013 and 2014 with a pronounced temporal variability in 2014 compared to rather small fluctuations in 2013. DOC quality characteristics were similar in autumns of both years, exhibiting an average range compared to the entire measurement period. During events in spring and autumn, $S_{275-295}$ and $SUVA_{254}$ remained at a constant level, indicating the export of DOC of similar composition.

Exported DOC loads (Table 1) peaked during high discharge events during spring and autumn and closely follow the hydrograph (Fig. S2). Accordingly, the CV of the load is closer to that of the discharge than to the CV of DOC (Table 1). Maximum DOC export was found during the discharge event on 27 April 2014 with rates of up to 18.6 g s$^{-1}$. Although events in drier summer months show stronger concentration fluctuations, exported loads remain low.

## 3.2 Correlation analysis

Table 2 gives an overview regarding correlations in the entire dataset. We use Spearman's rank ($r_s$) correlation to determine the direction and strength of relationships between variables. $C_{DOC}$ correlates strongest with $SUVA_{254}$, but $r_s$ between $C_{DOC}$ and $S_{275-295}$ and between $S_{275-295}$ and $SUVA_{254}$ is markedly smaller.

5 Correlations of $Q_{tot}$ with $S_{275-295}$ are stronger than $Q_{tot}$ with $SUVA_{254}$ and $C_{DOC}$, respectively. In comparison to $Q_{tot}$, correlations with $Q_{hf}$ are markedly higher for $C_{DOC}$ and $SUVA_{254}$, but lower for $S_{275-295}$. On the other hand, when relating $C_{DOC}$ and metrics of quality to the baseflow fraction of discharge ($Q_b$), $r_s$ is close to 0 for $C_{DOC}$ and $SUVA_{254}$, but 0.61 for $S_{275-295}$. $C_{DOC}$ and quality further correlate with antecedent discharge, temperature, discharge normalized temperature ($DNT_{30}$) and aridity index ($AI_{6, 14, 60}$). $C_{DOC}$ and $SUVA_{254}$ correlate best with $AI_6$, whereas $S_{275-295}$ correlate with $T_{30}$, $Q_{15}$, $Q_{30}$, $DNT_{30}$ and $AI_{60}$.

### 3.2.1 Event-scale analysis

High coefficients of determination ($R^2$) between $C_{DOC}$ and DOC quality metrics with $Q_{hf}$ and in the case of $S_{275-295}$ with $Q_b$ underline the prominent role of discharge and its different time scales for DOC variability. Consequently, quantifying DOC mobilization for a range of individual events may provide information for better understanding direction, shape and strength of C-Q relationships. The analysis of the response of $C_{DOC}$ and DOC quality to discharge events covers 44 % of the entire

time series. The relationship between $C_{DOC}$ and $Q_{tot}$ during events resembles a segmented slope in log-log space (Fig. S3a), similar to the C-Q behavior described by Moatar et al. (2017), which inhibits a proper parameterization by the usually applied simple power law regression. However, when detrending the discharge by baseflow subtraction, the resulting $C_{DOC}$-$Q_{hf}$ relationship is more linear in log-log space (Fig. S3b). This behavior occurs for the event-scale discharge variability of

the entire dataset. For DOC quality metrics $SUVA_{254}$ and $S_{275-295}$ we applied a similar model to predict the non-transformed independent variables:

$$Y = a \log(Q_{hf}) + b\, Q_b + z \tag{2}$$

where $Y$ is log($C_{DOC}$), $SUVA_{254}$ or $S_{275-295}$, resp.; $a$, $b$ are regression coefficients and $z$ is the intercept.

We applied Eq. (2) to 38 individual discharge events. The mean $R^2$ of all log($C_{DOC}$) models (one model for each discharge

event) is 0.84 ($\pm 0.15$). Respective mean $R^2$ values for $SUVA_{254}$ and $S_{275-295}$ were 0.83 ($\pm 0.14$) and 0.64 ($\pm 0.26$). Performance of the models is always better than a simple linear regression with log($Q_{tot}$) (mean $R^2$ for log($C_{DOC}$), $SUVA_{254}$ and $S_{275-295}$ is 0.76 ($\pm 0.16$), 0.70 ($\pm 0.15$) and 0.50 ($\pm 0.26$), respectively). $R^2$ of the models from Eq. (2) varies over time (Fig. 3). Dependent variables log($C_{DOC}$) and $SUVA_{254}$ show a similar behavior with maximum $R^2$ in autumn and winter and minimal $R^2$ values in spring and summer (Fig. 3a, b). $R^2$ of the $S_{275-295}$ models show a different and less consistent pattern with higher variability

between events than $C_{DOC}$ and $SUVA_{254}$ models (Fig. 3c). In comparison to $C_{DOC}$ and $SUVA_{254}$, $S_{275-295}$ values in winter and spring events have a systematically lower $R^2$.

Coefficients of $C_{DOC}$ and DOC quality models vary between the events (Fig. 3a - c). Coefficient $a$ (regression coefficient of $\log(Q_{hf})$) shows low but more systematic variations over time, represented by a smaller CV in comparison to $z$ and $b$ (mean $CV_a = 0.76$, mean $CV_z = 2.58$, mean $CV_b = 5.30$ of the $C_{DOC}$, $SUVA_{254}$ and $S_{275-295}$ models). High $a$ values indicate a stronger increase in $C_{DOC}$ and change in quality of DOC with an increase in $Q_{hf}$, whereas small $a$ values indicate only little change

with increasing $Q_{hf}$. All three models show a distinct change in $a$ from dry summer to autumn 2013. The summer months generally show the strongest variability in model coefficient, meaning that $C_{DOC}$ and DOC quality reacted strongly and more variable to the comparable small discharge events. Winter months in contrast show least variability in model coefficient $a$ indicating a more homogeneous reaction to discharge in this time of the year. Baseflow and intercept model coefficients $b$ and $z$ have a similar, less distinct, pattern for all three models with higher parameter variability in summer compared to the

other months.

### 3.2.2 Seasonal-scale analysis

A correlation analysis of the model coefficients $a$, $b$ and intercept $z$ was performed to identify the variables that explain their temporal dynamics (Table 3). More specifically, we aim to predict $a$, $b$ and $z$ by hydroclimatic conditions before and during

the event represented by the medians of $DNT_{30}$, different temporal aggregations of $AI$, $T$ and $Q$. Again, we rely on Spearman's rank correlation to characterize and quantify the relationships more independent of their shape. Intercept $z$ as well as coefficient $b$ (related to $Q_b$) do not show any correlation at p<0.001. Regression coefficient $a$ (related to $Q_{hf}$) shows good correlations (p<0.01) with $T_{15}$, $T_{30}$, $Q_{30}$, $AI_{60}$ and $DNT_{30}$ for all models. But median values of $DNT_{30}$ and $AI_{60}$ are the only variables which show highly significant correlations (p<0.001) with coefficient $a$ for $C_{DOC}$ as well as for the quality

metrics models. Strongest increase in $C_{DOC}$ within an event (high $a$) occurs when $AI_{60}$ is low and $DNT_{30}$ is high which translates into events during warm and dry low flow situations. On the other hand, during cold and wet high flow periods ($AI_{60}$ and $Q_b$ high, $DNT_{30}$ low) large events (high $Q_{hf}$) produce a smaller increase of $C_{DOC}$. This situation typically occurs during winter.

Based on the highest $r_s$ values in the correlation analysis for the event scale (Table 3), we selected $DNT_{30}$ and $AI_{60}$ as variables to explain seasonal variations in regression coefficient $a$. The results were used to build a regression model for all available data of $C_{DOC}$, $SUVA_{254}$ and $S_{275-295}$. We added to the model of Eq. (2) the seasonal-scale $AI_{60}$ and $DNT_{30}$. In addition we added those interactions for which VIF < 10 (Eq. (1)): $\log(Q_{hf}) \times Q_b$, $AI_{60} \times DNT_{30}$ and $DNT_{30} \times Q_b$. These two additions

allow the model to account for temporal changes in the relationships of $C_{DOC}$ and DOC quality with discharge. Note that we, again, rely on power law behavior of $C_{DOC}$ but logarithmic (semi-log) behavior for $SUVA_{254}$ and $S_{275-295}$ (above):

$$Y = z + a\ \log(Q_{hf}) + b\ Q_b + c\ AI_{60} + d\ DNT_{30} + i \tag{3}$$

where $Y$ represents one of the three dependent variables log($C_{DOC}$), $SUVA_{254}$ and $S_{275-295}$. $a$, $b$, $c$, $d$ are regression coefficients, $z$ is the intercept. $i$ indicates valid interaction terms (VIF < 10, Eq. (1)) log($Q_{hf}$)×$Q_b$, $AI_{60}$×$DNT_{30}$ and $DNT_{30}$×$Q_b$.

The results of the modelling are depicted in Table 4 and Fig. 4. A basic overview of all regression parameters and model statistics is given in Table S1. The $C_{DOC}$ model performs best, explaining most of the overall variance ($R^2$ = 0.72 ± 0.04 five-fold cross-validation prediction error), compared to the mean $R^2$ of 0.84 for modeling single events only. $SUVA_{254}$ and $S_{275-295}$ models explain similar parts (0.64 ± 0.2 and 0.65 ± 0.0) of the overall variance compared to the mean $R^2$ for the events of 0.83 and 0.64, respectively. All models generally explain both, seasonal and event-scale variability (Fig. 4, $R^2$ see Table S2), but towards small values, residuals of the DOC quality models tend to overestimate, whereas residuals of the $C_{DOC}$ model increase with increasing concentration (Fig. S4). Yet, 95% of the residuals lie within a range of 1.08 mg L$^{-1}$ and –0.90 mg L$^{-1}$, ± 0.44 L m$^{-1}$mg-C$^{-1}$ and ± 2.2 ×10$^{-3}$ nm$^{-1}$ for the $C_{DOC}$, $SUVA_{254}$ and $S_{275-295}$ models, respectively.

Inspection of models taking only event-scale predictors (log($Q_{hf}$), $Q_b$ and interaction) or only seasonal-scale predictors ($AI_{60}$, $DNT_{30}$ plus their interaction) into account reveals that both sets of variables can explain a comparable part of the total variance ($R^2$ event scale: 0.40, 0.36, 0.47; $R^2$ seasonal scale: 0.42, 0.36, 0.48 for the $C_{DOC}$, $SUVA_{254}$ and $S_{275-295}$ models, respectively). Yet, when only using seasonal-scale drivers ($AI_{60}$ and $DNT_{30}$ plus their interaction), the general trend but no event-type variability is reproduced in the model (Fig. 4). On the other hand, the pure discharge model does not reproduce baseflow and peak height well during the seasons.

For the complete $C_{DOC}$ and $SUVA_{254}$ model, seasonal-scale drivers $AI_{60}$ and $DNT_{30}$ plus their interaction $DNT_{30}$×$AI_{60}$ and event-scale driver log($Q_{hf}$) alone are the most important predictors, able to explain 68% of the total variance for $C_{DOC}$ and 54% for $SUVA_{254}$ compared to 72% and 64% of the respective complete models (Table 4). In contrast to the $C_{DOC}$ and $SUVA_{254}$ models, the interaction of seasonal-scale drivers ($DNT_{30}$×$AI_{60}$) barely influences the $R^2$ of the $S_{275-295}$ model, but it is rather $DNT_{30}$ plus the interaction of $DNT_{30}$×$Q_b$ and event-scale hydrological drivers log($Q_{hf}$) and $Q_b$ which alone can explain 54% of the variance compared to 65% of the complete model.

Interactions between $AI_{60}$ and $DNT_{30}$ play a crucial role in the $C_{DOC}$ and $SUVA_{254}$ models. There is a small negative effect of increasing soil wetness during low $DNT_{30}$ values and a small negative $DNT_{30}$ effect for dry soils. However, if exposed to increasing $AI_{60}$ values, the effect of medium and high $DNT_{30}$ values changes towards a positive interaction. Hence, when $AI_{60}$ is low and $DNT_{30}$ high, which typically occurs during the summer months (Fig. 2b) or vice versa in winter, interaction leads to the lowest mean $C_{DOC}$ and $SUVA_{254}$ values during non-precipitation periods (Fig. S5a, b). With medium $AI_{60}$ and $DNT_{30}$ values around autumn and spring, the interaction (Fig. S5c) has more positive influence on $C_{DOC}$ and $SUVA_{254}$ values, resulting in higher baseflow $C_{DOC}$ and $SUVA_{254}$ values. This interaction can thus represent the change of regression coefficient $a$ that was observed in the event analysis (Fig. 3). In comparison to the $C_{DOC}$ and $SUVA_{254}$ models, for the $S_{275-295}$ model the interaction of log($Q_{hf}$) with $Q_b$ has direct influence on the time variant regression coefficient $a$ and thus more influence on the $R^2$ (Table 4).

There is a positive effect of increasing $Q_b$ at low and medium $\log(Q_{hf})$ values and a positive $\log(Q_{hf})$ effect during low $Q_b$. However, the effect of $\log(Q_{hf})$ changes towards a negative interaction if exposed to increasing $Q_b$ so that $\log(Q_{hf})$ barely increases $S_{275-295}$ values during high $Q_b$ situations.

 **4 Discussion**

**4.1. Performance of event-scale and complete models**

Within one year, DOC concentration and quality dynamics fluctuate on event and seasonal scale. The regression models revealed that discharge had a different impact on observed DOC concentration than on observed DOC quality in the Rappbode stream at the seasonal scale (Fig. 3). We found that during summer initial $C_{DOC}$ was low during baseflow while large amounts of DOC were available to be exported from the riparian soils to the stream during events leading to high model coefficient $a$ (Fig. 3). Contrarily, the increase in concentration in winter is less pronounced (low model coefficient $a$, Fig. 3) because there is less DOC available to be washed out. Although the largest amounts of exportable DOC are to be expected at the end of the summer and in early autumn (Clark et al., 2005), $C_{DOC}$ and DOC quality changed most distinctly with the discharge components $Q_{hf}$ and $Q_b$ in the summer (Fig. 3). Unfortunately, there were no DOC measurements of the riparian soil water available which could further elucidate this discrepancy.

The regression models across the entire observed time series (section 3.2.2) utilize event-scale drivers $\log(Q_{hf})$ and $Q_b$ as well as more seasonally driven variables $AI_{60}$, $DNT_{30}$ and their interactions to explain DOC concentrations and quality variations. We are aware that predictions based on statistical relationships between predictors and DOC responses, which are outside the range of the calibration data (e.g. during extreme droughts and flooding) have to be treated with care. Furthermore, validity and sensitivity of the statistical relationships with the predictors does not account for long-term changes in biogeochemical and hydroclimatical factors but can influence DOC export behavior on its own. Other influences not regarded in this model are the occurrence of chemical compounds like nitrogen (Garcia-Pausas et al., 2008), sulphate, chloride or acid deposition (Futter and de Wit, 2008) which all can impact the available forms, stability and mineralization of carbon in soils. Studying the interactions of DOC with other elements could therefore be useful to add understanding to the actual mobilization and processing mechanisms. But since we measured DOC in the stream, we view DOC as an integrated response signal, already carrying all the information from processing and transformation up to abiotic removal in the riparian zone. Thus, we argue that hydroclimatic and discharge dynamics as chosen here, are a first order controls of the DOC dynamics in the stream, represented by a high correlation coefficient between hydroclimatic variables and DOC quantity and quality (Table 3) as well as an $R^2$ of 0.72 for the complete $C_{DOC}$ model. Also, the complete $C_{DOC}$ model represented well the observed cumulative DOC export with a Nash-Sutcliffe efficiency (NSE) of 0.998 throughout the year. Taken by themselves, seasonal-scale drivers ($DNT_{30} + AI_{60} + DNT_{30} \times AI_{60}$) were able to explain the same amount of $C_{DOC}$ variability than hydrological event-scale drivers ($Q_{hf} + Q_b + Q_{hf} \times Q_b$). But with an NSE of 0.979 cumulative modeled DOC export from

event-scale drivers resembled actual cumulative DOC export much better than seasonal-scale drivers alone (NSE = 0.783), indicating that predictors based on low frequency measurements alone are not able to explain DOC export as accurately as those derived from higher frequency measurements. The different export behavior obtained from DOC export modeling based on low versus high frequency measurements is most pronounced during events (Fig. S6), which, again highlights the importance of high frequency measurements.

We used an hourly resolution for modeling $C_{DOC}$ and DOC quality (~17,000 values in ~ 1 year). In a low frequency study, Köhler et al. (2009) took 470 stream water samples in 14 years (based on Köhler et al. (2008)). Consequently the DOC concentration variance, which was needed to be explained, shifted from a focus on seasonal scale and inter-annual variations in Köhler et al. (2009) towards high-frequent fluctuations on top of the seasonal-scale shifts and thus a more holistic perspective in the present study. In addition, Köhler et al. (2009) did not analyze the processes which are responsible for the shifts between the models, which had been independently set up for snow covered, melting and snow free periods.

Other studies took higher observational frequency into account and added DOC source characterization to better understand the mobilization dynamics: e.g. Broder et al. (2017) and Tunaley et al. (2016) examined event driven changes in DOC export in a headwater stream, based on highly-resolved (15 min to 3 hour frequency) events. Like in the present study, both found that antecedent wetness conditions and seasonality are related to DOC dynamics in streams. Both studies provided a qualitative and descriptive assessment only and concluded that a more specific understanding of how DOC gets exported from catchments (Tunaley et al., 2016) might become even more important with respect to future changes in the hydrologic regime due to climate change (Broder et al., 2017). We argue that we need a better quantitative understanding of hydrological and biogeochemical mechanisms and interactions based on time series of different key controlling variables covering all relevant process-scales in terms of resolution and length.

Several authors identified seasonality as an important driver for DOC dynamics (Ågren et al., 2007; Broder et al., 2017; Tunaley et al., 2016). However, the term "seasonality" is rather vague and often not clearly defined in terms of its impact on DOC export. This makes its use for a quantitative comparison between catchments and different climates difficult. Therefore we used a set of more easily identifiable, quantitative hydroclimatic variables instead, which reflect the general seasonal dynamics (Table 3) and at the same time allow for a better assessment of the dominant processes for DOC concentration and quality variations.

In summary, we used high-frequency measurements of hydroclimatic variables and their interactions as a proxy-representation for seasonality, which allows a more quantitative comparison to other catchments and a more in depth evaluation of the system.

## 4.2 Hydroclimatic classification

To estimate how event-scale and seasonal controls interact to produce the observed non-linear responses of DOC concentrations and quality in our study catchment, we can separate the observation period into three distinct hydroclimatic

states. These three discrete system states were chosen to highlight certain, typical scenarios out of a continuum of hydroclimatical conditions, which are based on the seasonal-scale predictors of the complete regression models (Fig. 5): 1) high $DNT_{30}$ and low $AI_{60}$, representing warm & dry situations mainly found in summer 2) moderate $DNT_{30}$ and $AI_{60}$, representing intermediate warm and wet situations, mainly found in spring and autumn and 3) low $DNT_{30}$ and high $AI_{60}$, representing cold & wet situations mainly found in winter. To synthesize our modelling results in terms of potential underlying mobilization processes, these three states were compared by looking at both event and non-event responses of DOC concentrations and quality during those states.

Daily mean $C_{DOC}$, $SUVA_{254}$ and $S_{275-295}$ values of 1.49 mg L$^{-1}$, 0.68 L m$^{-1}$ mg-C$^{-1}$ and 5.0 $\times 10^{-3}$ nm$^{-1}$ were minimal at the end of the drought in August 2013, when baseflow levels were low, whereas values of 4.14 mg L$^{-1}$, 4.05 L m$^{-1}$ mg-C$^{-1}$ and 15.8 $\times 10^{-3}$ nm$^{-1}$ were measured during phases with higher baseflow levels in the cold & wet state. $C_{DOC}$, $SUVA_{254}$ and $S_{275-295}$ values showed the strongest increase during warm & dry situations (Fig. 5) also indicated by highest slopes of regression coefficient $a$ (event-scale models, Fig. 3). Events during the intermediate state also showed elevated $C_{DOC}$, $SUVA_{254}$ and $S_{275-295}$ values, but in comparison to summer events at a decreased variance and range (Fig. 5). Changes due to events in cold & wet situations were small in range and variance. Variance and mean of $S_{275-295}$ were generally lower during warm & dry situations than during intermediate and cold & wet phases. Therefore we conclude that seasonal-scale hydroclimatic variance controls the overall variance of $S_{275-295}$, whereas $C_{DOC}$ and $SUVA_{254}$ are driven through event type variance.

## 4.3 Conceptual model of DOC mobilization from the riparian zone

The relationship between $AI_{60}$ and $DNT_{30}$ in combination with differences in DOC concentration and quality of the three states is of particular interest to support a mechanistic explanation for differing DOC export during events. Hence, these metrics can be utilized for conceptualizing DOC mobilization dynamics of seasonal-scale variations in $C_{DOC}$ and the observed quality-discharge dependencies (Fig. 6).

### 1) Warm & dry situations

Warm & dry situations are hydroclimatically defined by high temperatures and low mean discharge (high $DNT_{30}$), relatively dry soil conditions (low $AI_{60}$) as well as low baseflow levels, as typically found in summer when the Rappbode is fed mainly by deeper riparian groundwater. During baseflow conditions highly processed DOC enters the stream via the deeper groundwater flow paths (Broder et al., 2017). DOC in deeper groundwater usually has passed through multiple soil layers, its amount and its composition has been altered by sorption and biogeochemical processes (Inamdar et al., 2011; Kaiser and Kalbitz, 2012; Shen et al., 2015). Low $S_{275-295}$ values indicate high molecular weight of DOC with a dominance of terrestrial waters (Helms et al., 2008; Spencer et al., 2012) entering the stream during that time. Precipitation events can get buffered and retarded in the soils (low $Q_{hf}$) (state warm & dry, Fig. 6). Due to the soil type and generally high groundwater tables in our catchment, soil moisture can remain high, even when there was no rainfall for some time. Yet, lower water contents can increase the mineralization rate compared to (oxygen free) water-logged soils. However, Kalbitz et al. (2000) and citations

therein report a positive correlation between mineralization rate and DOC concentration of the soil solutions. In consequence, DOC production can be higher than mineralization in the unsaturated riparian zone environment (Kalbitz et al., 2000; Luke et al., 2007) leading to a net production of DOC. Hence, favorable conditions for the accumulation of DOC during non-event periods exist in the subsurface due to the lack of moving water in the topsoil, where the high temperatures

allow for (microbially driven) riparian DOC net production. To account for the positive balance between DOC removal mechanisms (mineralization, degradation) and DOC production in the riparian soil, we will use the term *net production* in the following.

We argue that the increase of $C_{DOC}$ and change of DOC quality with discharge events is due to the addition of a new, distinct DOC source, located in the shallow riparian soils and connected via transmissivity feedback and preferential flow paths (Fig.

S7). Since $C_{DOC}$ during non-event situations was very low (Fig. 5), higher DOC concentrations exported from the topsoils with different quality were able to override the low flow DOC signal towards a riparian zone signal. Respectively DOC quality during events changed markedly towards higher $SUVA_{254}$ values typical for higher aromaticity of the organic matter and associated to processed DOC (Hansen et al., 2016; Helms et al., 2008) and higher $S_{275-295}$ (but not as high as in cold & wet) indicating a *relative* increase in low molecular weight components in comparison to the low flow signal.

The (de)activation of an additional DOC source with changes in discharge could also explain the observed lower R² values in the event analysis during summer (Fig. 3), because in this situation, $C_{DOC}$ is not only driven by discharge but an addition of a differing DOC source that is not explained by the hydrological drivers of the event-scale models. The extend of this additional DOC source is determined by antecedent hydroclimatical conditions which favor DOC net production and thus indicated a sensitivity to biogeochemistry driven DOC export as found by Winterdahl et al. (2016) on top of a general

transport limited system (Zarnetske et al., 2018). Accordingly, event analysis showed the highest $C_{DOC}$ and DOC quality peaks and revealed the steepest $C_{DOC}$-$Q_{hf}$ and quality-$Q_{hf}$ relations in summer. After the event, $C_{DOC}$ and DOC quality metrics gradually drop back to the baseflow signal.

In contrast to our findings, Raeke et al. (2017) found higher molecular weight molecules at elevated discharge in three temperate catchments (including the one studied here). However, they used grab samples from different hydroclimatic

situations and streams, thus potentially masking the event-scale dynamics of DOC mobilization as revealed in the current study. Also the comparability between spectrophotometry and high resolution mass spectrometry is questionable for DOC in general (Chen et al., 2016). But also the magnitude of in-stream processing and biodegradation could further influence DOC composition and hence $SUVA_{254}$ and $S_{275-295}$ measurements in stream water (Bernal et al., 2018; Hansen et al., 2016). However, Creed et al. (2015), Nimick et al. (2011), stated that headwaters in general are dominated by allochthonous carbon

with the role of in-stream processing increasing with stream order. Also the role of in-stream processing at mean residence times below one day (which holds for our study site, 2km downstream of the spring) was found to be minor (Kaplan et al., 2008; Köhler et al., 2002). Note that the wide riparian zone (several tens of meters) in our catchment consists to large parts of a flood plain, leaving only little possibility for leaf litter falling directly into the stream. Therefore, in-stream decomposition and leaf litter in the stream are likely to be of minor importance on our experimental site.

2) Intermediate state

Intermediate $DNT_{30}$ and $AI_{60}$ conditions are defined by moderate temperatures and discharge (medium $DNT_{30}$), precipitation and evapotranspiration (medium $AI_{60}$) which results in higher baseflow levels as compared to warm & dry conditions. Strong precipitation events translate into a distinct discharge signal (high $Q_{hf}$) (state intermediate, Fig. 6). Conditions for the accumulation of DOC during non-event periods are less favorable due to colder temperatures than warm & dry, decreasing the riparian DOC net production. During baseflow conditions some of the riparian DOC pools are already activated due to a higher groundwater table. This mixing of riparian and deeper groundwater DOC pools translates into intermediate values of concentration and quality parameters, even under non-event conditions.

In case precipitation increases discharge, the DOC signal changes both concentration and quality. This process happens faster than during the warm & dry situation, since antecedent wet conditions facilitate DOC mobilization from riparian soils. Hence the temporal shift between DOC and discharge peak diminishes, resulting in higher R² values during events (Fig. 3). There was no exhaustion of the exportable DOC by consecutive events although there is less DOC production paired with more effective export mechanisms, highlighting the large store of DOC in the comparably small riparian zone (Ledesma et al., 2015). The intermediate situation averages multiple situations (transition states in autumn and spring) and thus does not have the character and clarity of the endmembers. Similar quality signals indicate the same process and location of source zone activation in autumn 2013 and 2014. However, concentration peaks developed differently, suggesting that the conditions for antecedent DOC storage and export during preceding phases were different. E.g., there were only little mobilization and storage limitations during intermediate $DNT_{30}$ and $AI_{60}$ levels in spring 2014, which translated into pronounced DOC loads exported during events. However, DOC quality, especially $S_{275-295}$ barely changed during these events. Elevated temperatures during this period cause a warming of riparian topsoil, which are rich in organic matter, and hence an increase in biological processing and DOC production. Declining, still high baseflow levels and soil moisture lead to increased DOC production and export during these events.

3) Cold & wet situations

Cold & wet situations, mainly found in winter, are defined by low temperatures and high mean discharge (low $DNT_{30}$), humid conditions (high $AI_{60}$) as well as high baseflow levels (state cold & wet, Fig. 6). Generally low $C_{DOC}$ values indicate that less DOC mass is available in relation to the generated runoff in the riparian zone in comparison to the warm & dry situations. Unfavorable conditions for the net production of DOC during non-event periods exist in the topsoil, where the low temperature impairs riparian DOC production. Accordingly, low $SUVA_{254}$ and high $S_{275-295}$ values were observed during that period, indicating relatively higher amount of low molecular weight compounds due to reduced DOC processing. Furthermore, high base flow levels lead to a good hydrological connectivity of DOC sources to the stream during non-event situations.

Precipitation events result in small slopes of the $C_{DOC}$ and quality-$Q_{hf}$ relationships. Dilution due to the impermeability of the frozen soil surface (Laudon et al., 2007) is likely to occur under prolonged periods of temperatures below zero. Since

riparian DOC pools are already connected to the stream, we attribute the small shift in DOC quality and $C_{DOC}$ during events to a shift of the contribution (hydrological connection) of DOC source areas with similar DOC quality, rather than to the activation of new, differing DOC pools. The first order hydrological forcing under largely saturated soil conditions thus could explain the high R² but low regression coefficient *a* of the event-scale models of $C_{DOC}$ and $SUVA_{254}$ (Fig. 3) in the cold
& wet state. On the other hand, a dominance of hydrological forcing also implies little influence of antecedent biogeochemical conditions during this state (Winterdahl et al., 2016). In contrast to $C_{DOC}$ and $SUVA_{254}$, R² of $S_{275-295}$ drops during the cold & wet situation, indicating a decoupling from hydrologic forcing. The dominant hydrological state could be able to leach differing DOC from the riparian zone by shifts in physicochemical equilibria (Shen et al., 2015) thereby forming the corresponding quality. However, this finding needs further research.

The same observations of $C_{DOC}$ and quality interaction during winter and spring (low DOC variance in winter, still low quality variance but strong $C_{DOC}$ fluctuations in spring) were made in 2013. But due to the lack of weather data (the weather station was deployed two months after the sensor deployment which inhibited derivation of *AI* and *DNT* for this period), no further statements can be made for this period (Fig. 2).

## 5 Conclusions

Seasonal- and event-scale DOC quantity and quality dynamics in headwater streams are dominantly controlled by the dynamic interplay between event-scale hydrological mobilization and transport (delivery to the stream) and inter-event and seasonal biogeochemical processing (exportable DOC pools) in the riparian zone. Observing DOC concentration and quality, together with hydroclimatical factors, at high frequency resolves dynamics at the temporal scale of the underlying hydrological and biogeochemical processes, which is unattainable with standard grab-sample monitoring. This allows for an
improved, in-depth assessment of DOC export mechanisms as joint measurements of DOC quantity and quality give additional insights into source locations in the riparian zone, DOC processing and mobilization.

Observed DOC concentration, $SUVA_{254}$ and $S_{275-295}$ averaged at 4.06 mg L$^{-1}$, 3.93 L m$^{-1}$ mg-C$^{-1}$ and 13.59 × 10$^{-3}$ nm$^{-1}$, respectively, but were found to be highly variable in time. The analysis of event-scale variability revealed clear seasonal-scale shifts of the role of discharge in shaping DOC quantity and quality. Overall, the temporal dynamics of DOC
concentration and quality can be explained by a few key controlling hydrological variables, which characterize instantaneous discharge, and hydroclimatic metrics, which define the conditions prior to the event.

The hydrological variables ($Q_{hf}$ and $Q_b$) were able to explain 40%, 36% and 47% of the overall variability of $C_{DOC}$, $SUVA_{254}$ and $S_{275-295}$ and play a crucial role for modeling DOC export. In comparison, seasonal-scale variables ($AI_{60}$ and $DNT_{30}$) alone are able to explain similar percentages (42%, 36%, 48% for $C_{DOC}$, $SUVA_{254}$, $S_{275-295}$) of the overall variability of DOC
quantity and quality, but lack in adequately predicting exported DOC loads. Combining both sets of variables, as done in this study, significantly increases the predictive capacity of the overall models (72%, 64%, 65% for $C_{DOC}$, $SUVA_{254}$, $S_{275-295}$). Evaluation of the developed statistical models also highlights the importance of interactions between the seasonal-scale

antecedent predictors $AI_{60}$ and $DNT_{30}$ on DOC concentration and quality dynamics. $AI_{60}$ describes the potential for mobilizing DOC in riparian soils, whereas $DNT_{30}$ describes the changes in DOC storage by looking at the relationship of DOC production and prior mean export from riparian soils. Hence, the relationship between $AI_{60}$ and $DNT_{30}$ describes the potential for export DOC from riparian soils and allows us to conceptualize DOC exports under differing hydroclimatical conditions. We found that cold & wet situations ($AI_{60}$ high, $DNT_{30}$ low) are not mobilization limited (high mobilization potential due to wet soils and high baseflow levels) but limited in production and processing (due to low temperatures). High hydrological connectivity leads to low $C_{DOC}$ when the DOC net production is low compared to the DOC export. Here, events do not change the quality signature of the DOC in the stream, since all riparian DOC sources had already been connected to the stream before. In contrast, we interpret warm & dry conditions ($AI_{60}$ low, $DNT_{30}$ high) as mainly mobilization-limited situations (dryer soils, low baseflow levels). High DOC net production rates (high temperatures) and low hydrological connectivity lead to an accumulation of DOC in the upper soil layers of the riparian zone during non-event situations. Under those baseflow conditions low concentrations of highly processed DOC are exported from deeper soil layers to the stream. Overall, DOC quality varies the most during such warmer dry periods, because events change the signature of DOC quality in the stream water by adding freshly processed DOC from upper riparian DOC sources to the older more intensely processed DOC from the underlying base flow signature.

The findings reported and analyzed here provide a mechanistic explanation of the seasonally changing characteristics of DOC-discharge relationships and therefore can be utilized to infer the spatio-temporal dynamics of DOC origin in riparian zones from the DOC dynamics of headwater streams.

Our interpretation is based on the integrated signal of DOC concentration and quality measured in the stream. Accordingly, it remains partially unresolved, which explicit processes in the riparian zone are responsible for the measured and conceptualized DOC dynamics in the Rappbode stream. Further research in the riparian zone with its shallow groundwater dynamics is necessary to fully mechanistically explain the explicit spatio-temporal mobilization patterns as well as to identify appropriate molecular markers that can be used to trace DOC from riparian source zones into the stream in order to better understand DOC mobilization processes.

The study demonstrates the considerable value of continuous high-frequency measurements of DOC quality and quantity and their key controlling variables in quantitatively unraveling DOC mobilization in the riparian zone. We believe our approach allows long-term DOC monitoring with a manageable allocation of time and resources as well as a better comparability between catchments of different seasonal characteristics. This study highlights the dependency of DOC export on hydroclimatic factors. Potential impacts of climate change on the amount and quality of exported DOC are therefore likely and should be further investigated.

*Data availability*. All data sets used in this synthesis are publicly available via the link: https://doi.org/10.4211/hs.e0e6fbc0571149b79b1e75fa44d5c4ab.

*Author contributions*. JF, OL, AM and GdR planned and designed the research. MO carried out parts of the field work and conducted a first version of data processing and analysis. BW did the statistical analysis and wrote the paper with contributions from all co-authors.

*Competing interests*. The authors declare that they have no conflict of interest.

*Acknowledgements*. Acquisition of data was financially supported by the Federal Ministry of Education and Research Germany [grant number BMBF, 02WT1290A]. Special thanks to Toralf Keller for excellent and steady field work as well as
to Wolf von Tümpling for the support in the laboratory.

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

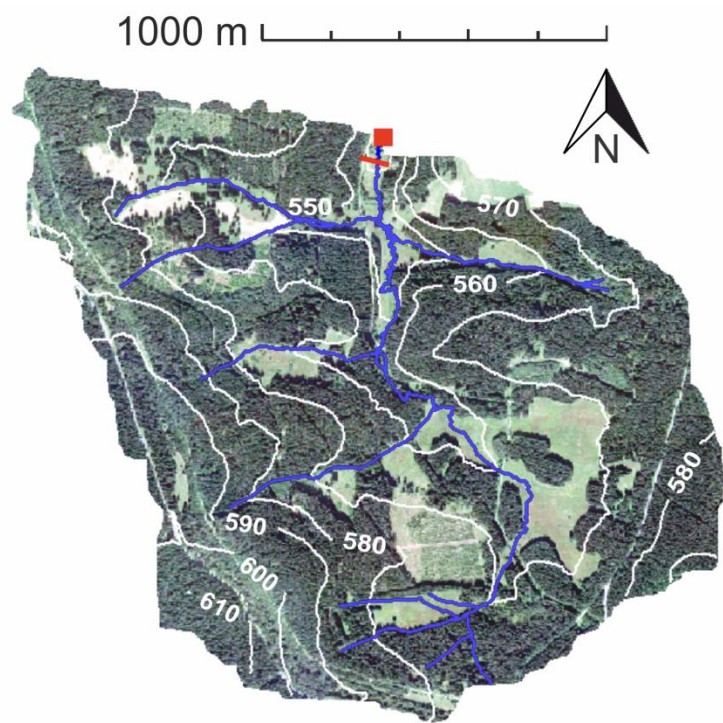

**Fig. 1: Topography of the Rappbode catchment with the UV-Vis and discharge measurements at the outlet (red square). Transect for soil samples indicated by red line.**

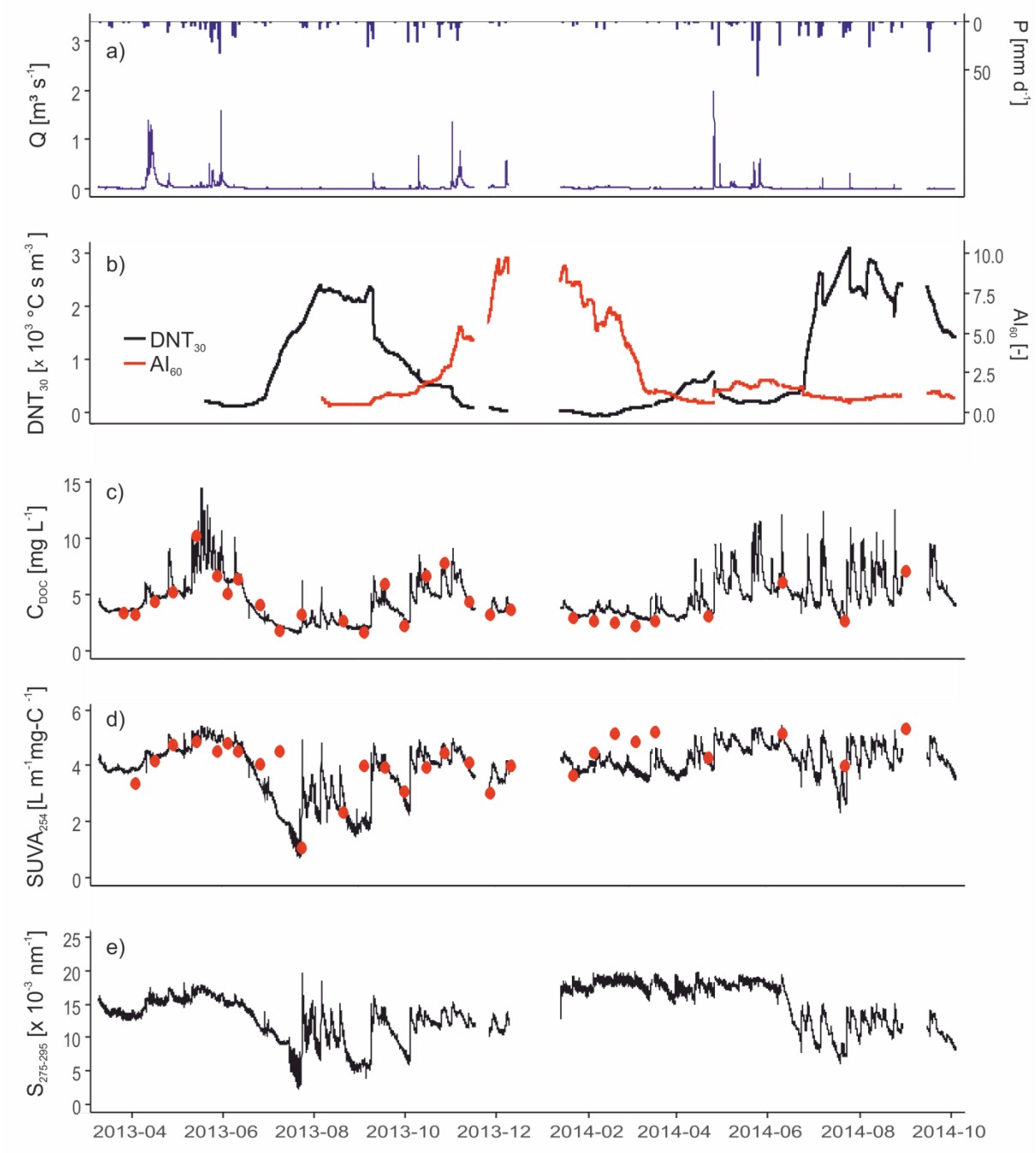

**Fig. 2: (a)** Precipitation and discharge, **(b)** antecedent hydro meteorological conditions, **(c)** $C_{DOC}$, **(d)** $SUVA_{254}$ and **(e)** $S_{275-295}$ over the entire measurement period. $C_{DOC}$ in (c) was fitted with PLSR to the measured grab samples (red dots). Grab samples (red dots) in the $SUVA_{254}$ values (d) were just used for validation.

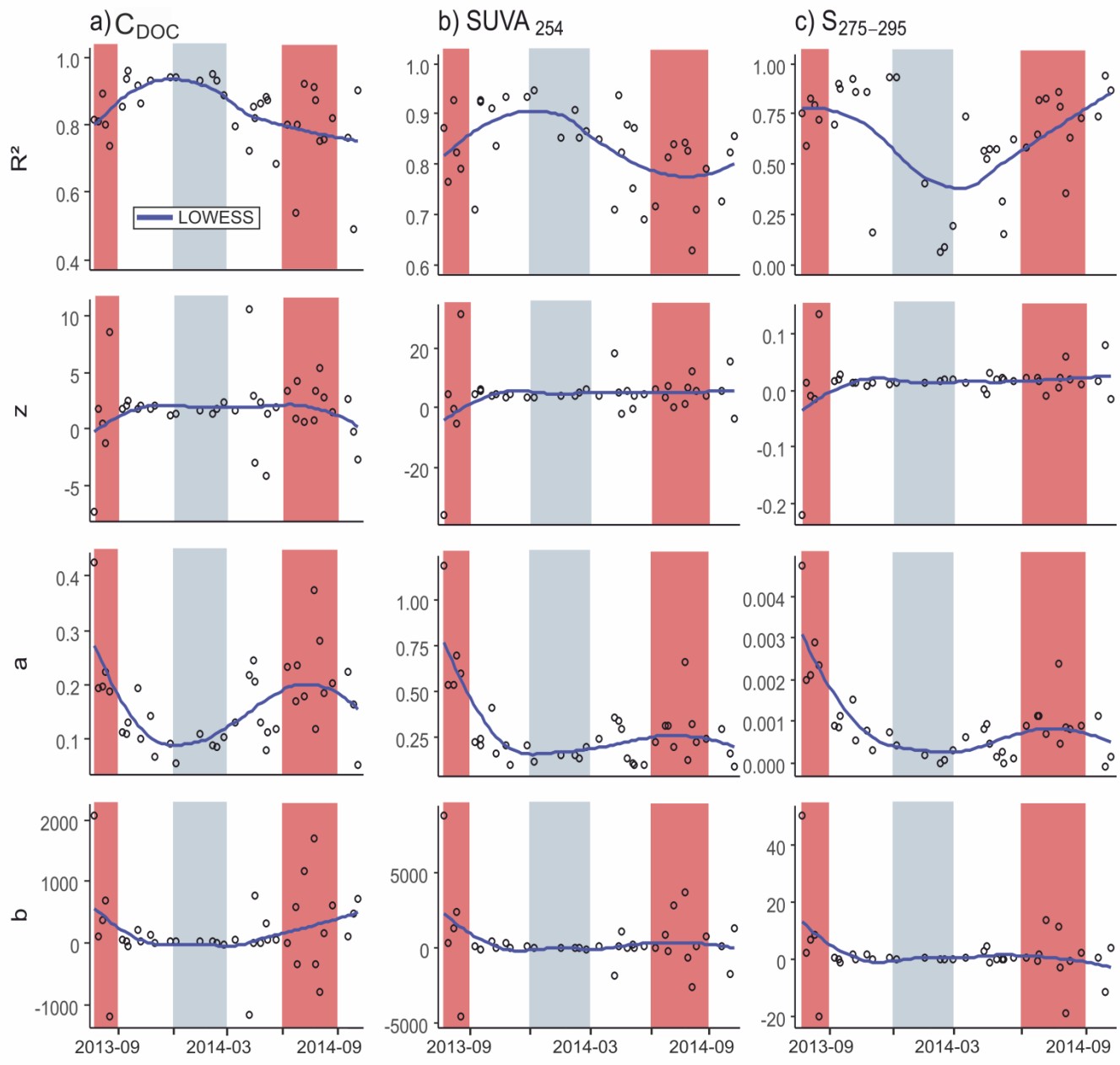

Fig. 3: R², intercept *z* and regression coefficients *a* and *b* of the model predictors log ($Q_{hf}$) and $Q_b$ in Eq. (2) of all 38 events plotted against time. The headings in the top of the figure indicate which variable was represented by Y in Eq. (2). Blue lines indicate the locally weighted scatterplot smoothing (LOWESS), background colors indicate seasons (grey = winter, red= summer, white = autumn and spring). Note the different scales of the y-axes.

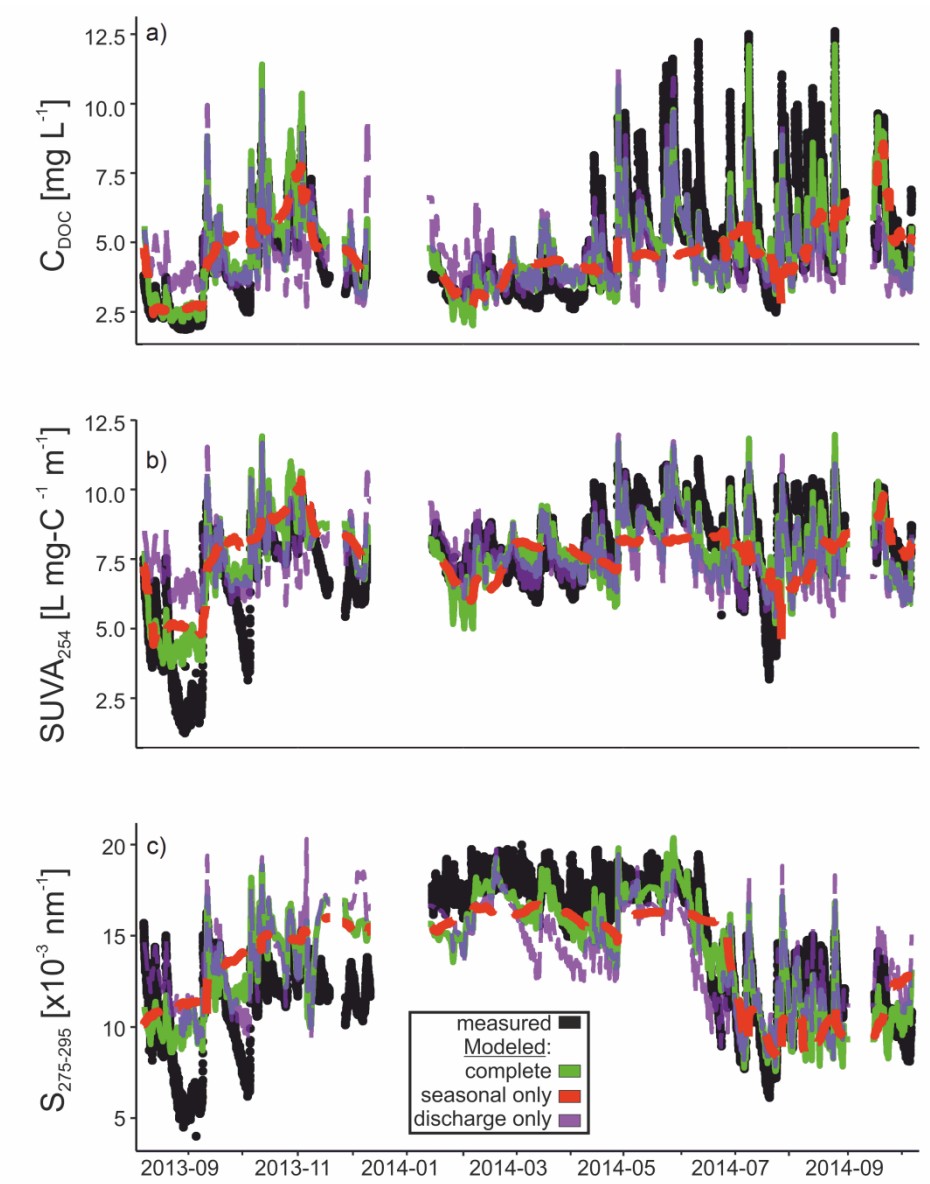

**Fig. 4: Comparison between measured (black) and multiple regression models of the complete predictors (green) as given by Eq. (3), only seasonal predictors $AI_{60}$ and $DNT_{30}$ plus their interaction (red) and only discharge predictors $\log(Q_{hf})$ and $Q_b$ plus their interaction (purple) for (a) $C_{DOC}$, (b) $SUVA_{254}$ and (c) $S_{275-295}$ values. Complete and discharge only model were smoothed (5 hourly) for better visualization.**

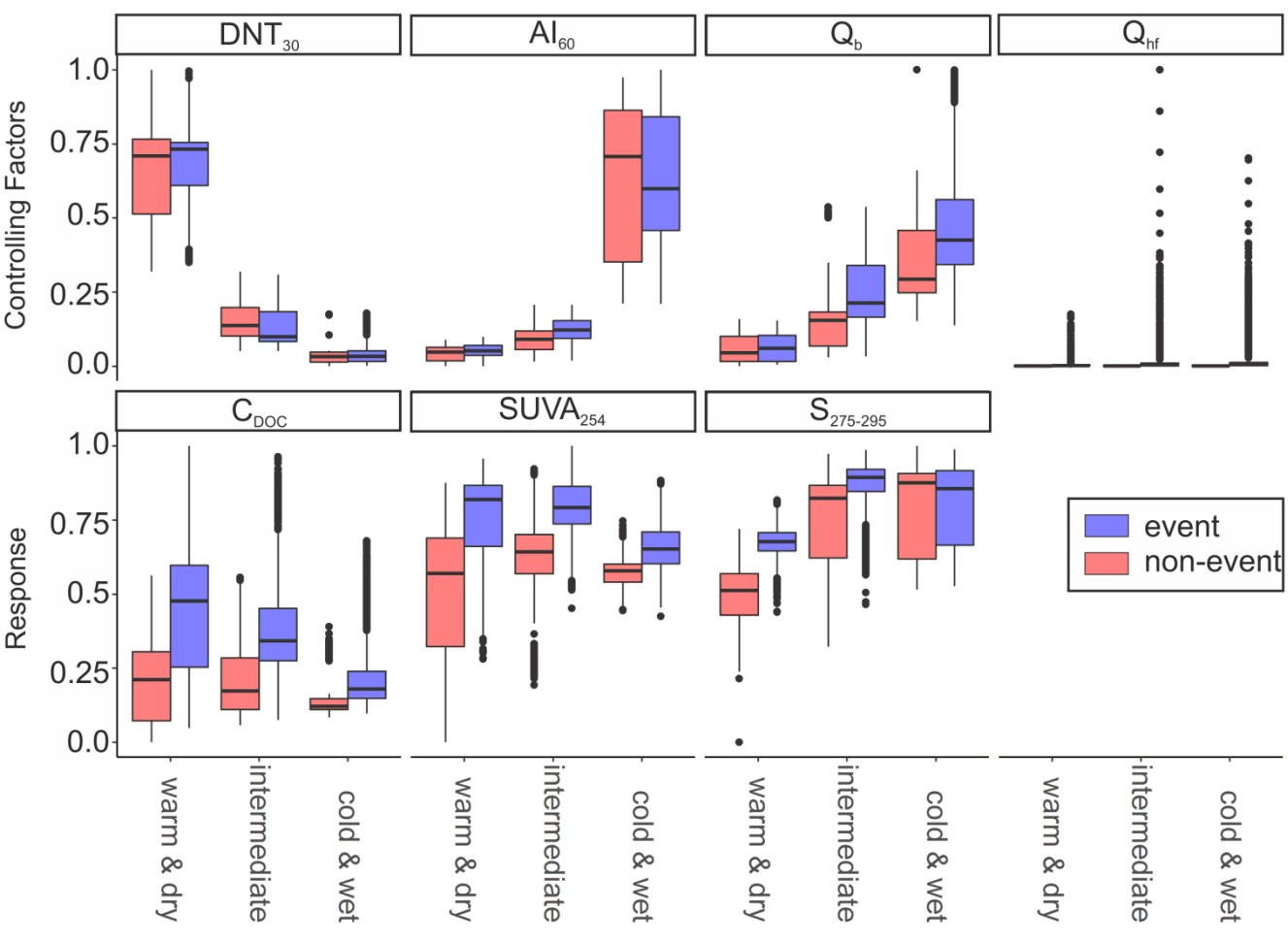

**Fig. 5:** Box-plots of hydroclimatic variables (controlling factors) and DOC quantity and quality metrics (response) classified into three hydroclimatic states: 1) warm & dry, 2) intermediate, 3) cold & wet. Red color indicates non-event situations, purple color event situations during the according states. Variables were rescaled for better illustration. Particular median $C_{DOC}$ values during non-event situations were 4.13 mg $L^{-1}$, 3.72 mg $L^{-1}$ and 3.16 mg $L^{-1}$ for the warm & dry, intermediate and cold & wet state, respectively. Both warm & dry and intermediate state differ highly significant (Kruskal-Wallis test, $p < 0.001$) from the cold & wet state.

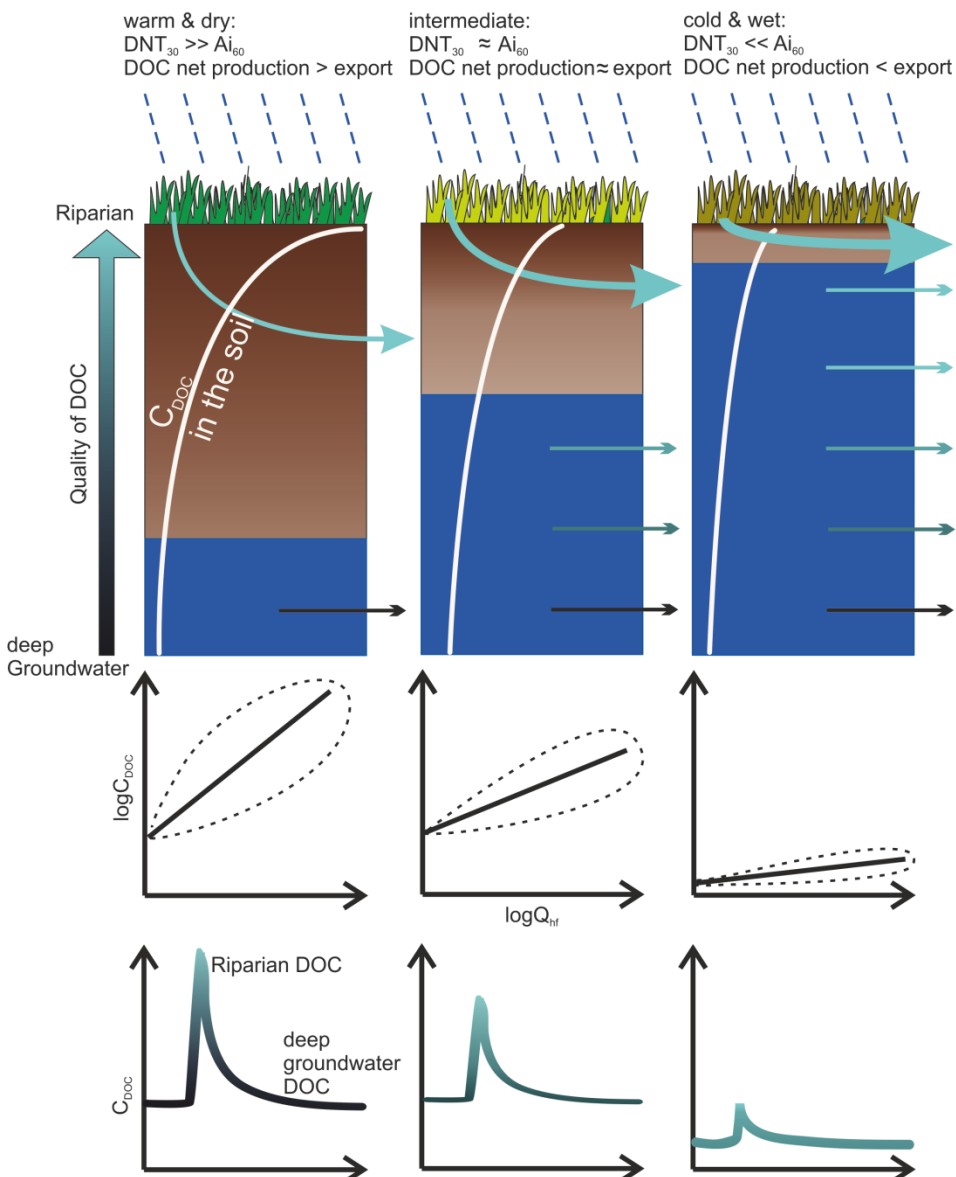

**Fig. 6: Conceptual model of riparian DOC export from precipitation during the three hydroclimatic states: warm & dry, intermediate, cold & wet. Depth of the soil column is around 0.5 m. Seasonal-scale variations of $C_{DOC}$ in the soil solutions (summer vs. winter) were e.g. discussed in Kalbitz et al. (2000).Changing combinations between $SUVA_{254}$ and $S_{275-295}$ values are described as more groundwater influenced (black) and more riparian influenced (green) DOC quality. Arrows indicate the export of DOC; colors of the arrows refer to the respective DOC quality. Panels in the middle row show the relation between $C_{DOC}$ and $Q_{hf}$ during the three representative situations. Dashed lines indicate the "dispersion" of the point cloud (according R²) during the events. Panels in the bottom line indicate the change of $C_{DOC}$ during an event. Corresponding changes of colors indicates more groundwater influenced (black) and more riparian influenced (green) DOC quality. Baseflow levels under cold & wet conditions are usually higher than baseflow levels during the warm and dry phase (see Fig. 5). Thus, during the cold and wet situation, higher layers of soil, more enriched in DOC get activated, but at the same time, there is also a tradeoff between amount of water and available DOC in the respective soil layers which can account for lower overall DOC concentrations.**

**Table 1: Descriptive statistics of DOC and hydroclimatic variables. N refers to number of measurements, St.Dev. – standard deviation, Min – minimum of the measurements, Max – maximum of the measurements and CV – coefficient of variation. Class shows if the variable was utilized as response (r) or predictor (p) in statistical models.**

| Variable | Description | Class | N | Mean | St. Dev. | Min | Max | Median | CV |
|---|---|---|---|---|---|---|---|---|---|
| $C_{DOC}$ [mg L$^{-1}$] | DOC concentration | r | 42,427 | 4.60 | 1.94 | 1.49 | 13.05 | 4.24 | 0.42 |
| $SUVA_{254}$ [L m$^{-1}$ mg-C$^{-1}$] | Specific UV absorbance at 254 nm | r | 42,427 | 3.93 | 0.89 | 0.68 | 5.44 | 4.08 | 0.23 |
| $S_{275-295}$ [× 10$^{-3}$ nm$^{-1}$] | Spectral slope between 275 nm and 295 nm | r | 42,421 | 13.59 | 3.76 | 2.44 | 19.98 | 13.42 | 0.28 |
| $Q_{tot}$ [m$^3$ s$^{-1}$] | Total discharge | - | 42,427 | 0.03 | 0.07 | 0.002 | 1.98 | 0.01 | 2.81 |
| *Specific $Q_{tot}$ [mm]* | Specific total discharge | - | 42,427 | 1.16 | 2.71 | 0.078 | 76.74 | 0.38 | 2.81 |
| $Q_{hf}$ [m$^3$ s$^{-1}$] | High-frequency quick flow | p | 39,371 [a] | 0.02 | 0.07 | 0.0001 | 1.97 | 0.002 | 4.51 |
| $Q_b$ [m$^3$ s$^{-1}$] | Low-frequency baseflow | p | 41,516 [a] | 0.01 | 0.01 | 0.002 | 0.06 | 0.007 | 0.91 |
| $P$ [mm d$^{-1}$] | Precipitation | - | 42,427 | 2.21 | 5.62 | 0.00 | 55.50 | 0.00 | 2.55 |
| $T$ [°C] | Air temperature | - | 42,427 | 9.20 | 6.96 | -11.75 | 31.77 | 9.15 | 0.76 |
| $ET_P$ [mm d$^{-1}$] | Potential evapotranspiration | - | 20,344 | 3.01 | 4.99 | 0.00 | 25.98 | 0.35 | 1.66 |
| $AI_{60}$ | Aridity Index of the last 60 days | p | 17,482 | 2.73 | 2.72 | 0.43 | 11.33 | 1.43 | 1.00 |
| $DNT_{30}$ [°C s m$^{-3}$] | Discharge normalized temperature of the last 30 days | p | 42,427 | 921.37 | 919.56 | -66.20 | 3,095.86 | 501.27 | 1.00 |
| DOC export [g s$^{-1}$] | DOC export | - | 42,427 | 0.17 | 0.67 | 0.005 | 18.63 | 0.04 | 3.88 |

[a] N of $Q_b$ and $Q_{hf}$ differs from $Q_{tot}$ due to the applied filtering method for baseflow separation.

**Table 2: Spearman's rho ($r_s$) of possible controlling variables over the entire observation period. Only complete cases were used (n = 17,082). All correlations are highly significant (p < 0.001) because of the large sample size, $r_s$ with absolute values larger 0.6 printed in bold for better readability. Numerical subscripts of *T, Q, AI, and DNT* indicate how many preceding days were aggregated.**

| | $SUVA_{254}$ | $S_{275\_295}$ | $T$ | $T_{15}$ | $T_{30}$ | $Q_{15}$ | $Q_{30}$ | $AI_6$ | $AI_{14}$ | $AI_{60}$ | $DNT_{30}$ | $Q_{tot}$ | $Q_{hf}$ | $Q_b$ |
|---|---|---|---|---|---|---|---|---|---|---|---|---|---|---|
| $C_{DOC}$ | **0.91** | 0.18 | 0.23 | 0.30 | 0.25 | 0.10 | 0.03 | 0.46 | 0.29 | 0.11 | 0.16 | 0.22 | 0.49 | -0.08 |
| $SUVA_{254}$ | | 0.50 | 0.13 | 0.13 | 0.05 | 0.22 | 0.17 | 0.44 | 0.26 | 0.18 | -0.05 | 0.37 | 0.59 | 0.08 |
| $S_{275-295}$ | | | -0.32 | -0.53 | **-0.63** | 0.58 | 0.56 | 0.20 | 0.22 | 0.47 | **-0.66** | **0.67** | 0.57 | **0.61** |
| $T$ | | | | **0.70** | 0.68 | -0.46 | -0.51 | -0.21 | -0.35 | -0.56 | **0.64** | -0.48 | -0.22 | **-0.61** |
| $T_{15}$ | | | | | **0.96** | **-0.60** | **-0.64** | -0.17 | -0.39 | **-0.71** | **0.85** | **-0.63** | -0.31 | **-0.79** |
| $T_{30}$ | | | | | | **-0.65** | **-0.68** | -0.15 | -0.35 | **-0.71** | **0.89** | **-0.66** | -0.34 | **-0.81** |
| $Q_{15}$ | | | | | | | **0.87** | 0.33 | **0.66** | **0.76** | **-0.80** | **0.80** | 0.57 | **0.86** |
| $Q_{30}$ | | | | | | | | 0.19 | 0.45 | **0.81** | **-0.89** | **0.71** | 0.49 | **0.79** |
| $AI_6$ | | | | | | | | | **0.67** | 0.33 | -0.18 | 0.53 | **0.60** | 0.37 |
| $AI_{14}$ | | | | | | | | | | **0.62** | -0.43 | **0.64** | 0.56 | **0.60** |
| $AI_{60}$ | | | | | | | | | | | **-0.86** | **0.67** | 0.47 | **0.73** |
| $DNT_{30}$ | | | | | | | | | | | | **-0.73** | -0.44 | **-0.86** |
| $Q_{tot}$ | | | | | | | | | | | | | **0.84** | **0.87** |
| $Q_{hf}$ | | | | | | | | | | | | | | 0.56 |

**Table 3: Spearman's rho ($r_s$) of the 38 $C_{DOC}$, $SUVA_{254}$ and $S_{275-295}$ model coefficients with hydroclimatic variables. Asterisks indicate p-values (*** - <0.001, ** - <0.01, * - <0.05), $r_s$ with absolute values larger 0.6 printed in bold.**

| Model Parameters | $T_{15}$ | $T_{30}$ | $Q_{15}$ | $Q_{30}$ | $AI_6$ | $AI_{14}$ | $AI_{60}$ | $DNT_{30}$ | $Q_{hf}$ | $Q_b$ |
|---|---|---|---|---|---|---|---|---|---|---|
| $z\ (C_{DOC})$ | 0.05 | 0.05 | 0.02 | -0.02 | 0.05 | 0.07 | -0.09 | 0.03 | 0.15 | -0.12 |
| $a\ (C_{DOC})$ | 0.55 *** | 0.52 *** | -0.48 ** | -0.43 ** | -0.52 ** | **-0.65** *** | **-0.66** *** | **0.63** *** | -0.55 *** | **-0.71** *** |
| $b\ (C_{DOC})$ | 0.25 | 0.25 | -0.31 | -0.31 | -0.19 | -0.33 * | -0.15 | 0.32 | -0.38 * | -0.25 |
| $z\ (SUVA_{254})$ | 0.07 | 0.06 | 0.04 | -0.06 | -0.10 | 0.04 | -0.10 | 0.04 | 0.01 | -0.09 |
| $a\ (SUVA_{254})$ | 0.50 ** | 0.51 ** | -0.50 ** | -0.40 * | -0.42 ** | -0.56 *** | **-0.64** *** | 0.58 *** | -0.54 *** | **-0.60** *** |
| $b\ (SUVA_{254})$ | 0.21 | 0.18 | -0.32 | -0.22 | -0.10 | -0.34 * | -0.14 | 0.25 | -0.29 | -0.23 |
| $z\ (S_{275-295})$ | 0.00 | -0.02 | 0.21 | 0.11 | -0.09 | 0.23 | 0.04 | -0.10 | -0.02 | 0.07 |
| $a\ (S_{275-295})$ | **0.62** *** | **0.63** *** | -0.54 *** | -0.41 * | -0.28 | -0.47 ** | -0.56 *** | **0.62** *** | -0.47 ** | **-0.64** *** |
| $b\ (S_{275-295})$ | 0.13 | 0.11 | -0.31 | -0.18 | -0.12 | -0.45 ** | -0.14 | 0.19 | -0.20 | -0.24 |

**Table 4: Evaluation of the whole data set model by dropping the least influencing variable according to AIC, starting from the complete models (Eq. (3)).**

| $C_{DOC}$ model | $R^2_{CDOC}$ | $SUVA_{254}$ model | $R^2_{SUVA254}$ | $S_{275-295}$ model | $R^2_{S275-295}$ |
|---|---|---|---|---|---|
| Complete | 0.72 | Complete | 0.64 | Complete | 0.65 |
| $-\log(Q_{hf})\times Q_b$ | 0.71 | $-DNT_{30}\times Q_b$ | 0.60 | $-AI_{60}\times DNT_{30}$ | 0.65 |
| $-DNT_{30}\times Q_b$ | 0.69 | $-\log(Q_{hf})\times Q_b$ | 0.56 | $-\log(Q_{hf})\times Q_b$ | 0.56 |
| $-Q_b$ | 0.68 | $-Q_b$ | 0.54 | $-AI_{60}$ | 0.54 |
| $-\log(Q_{hf})$ | 0.42 | $-AI_{60}\times DNT_{30}$ | 0.35 | $-DNT_{30}\times Q_b$ | 0.53 |
| $-AI_{60}\times DNT_{30}$ | 0.02 | $-DNT_{30}$ | 0.33 | $-Q_b$ | 0.51 |
| $-DNT_{30}$ | 0.02 | $-AI_{60}$ | 0.31 | $-DNT_{30}$ | 0.23 |
| $-AI_{60}$ | 0 | $-\log(Q_{hf})$ | 0 | $-\log(Q_{hf})$ | 0 |