# Peer review of "High frequency measurements explain quantity and quality of dissolved organic carbon mobilization in a headwater catchment"

_Biogeosciences, 2019_

## Referee Comment (RC1) · Anonymous Referee #1 · 23 May 2019

Werner et al. measured high frequency DOC concentrations and DOC quality-related optical parameters in a headwater catchment in Germany over a 1.5-year period. They linked these data with hydroclimatic variables in order to understand the mobilization process of DOC from the riparian zone at event and seasonal scales using an interesting statistical approach. The authors conclude that hydroclimatic factors alone explain most of the stream DOC quantity and quality dynamics, and propose a conceptual model of DOC mobilization from riparian zones characterized by three hydroclimatic states, namely (i) warm and dry, (ii) intermediate, and (iii) cold and wet. The use of high-frequency measurements obtained with recent sensor techniques has opened up the door to a better understanding of catchment processes, and seeing this linked with

the acknowledged critical role of riparian zones for catchment biogeochemistry is inspiring and of great value (although actual measurements of soil wetness or soil water chemistry in the riparian zone would be helpful).

The paper is well written, well-structured, and easy to follow; which is always appreciated in the review process. I am not an expert on some of the statistical analyses performed but they seem to be meaningful for the purpose of the paper and correctly applied. Interesting results are presented and a thought-provoking conceptual model proposed. Despite this, I think there is still some work to be done before this study can be accepted for publication in Biogeosciences. Particularly, below I challenge some of the interpretations related to the conceptual model that the authors propose. This, together with all the other questions, comments, and suggestions below, need to be addressed by the authors before a more positive recommendation.

General comments

My main concern is that, in my opinion, some important processes for the context of the present study, and particularly for the conceptual model proposed, are ignored. Specifically I refer to:

Mineralization (decomposition/respiration). When discussing DOC accumulation in the riparian zone, besides talking about production versus lateral mobilization one needs to account for mineralization, which is another way in which DOC can be lost. For example, the authors claim that warm and dry conditions are optimal for DOC accumulation because of increased production rates and low hydrological connectivity but these situations can also favour high oxygen supply and thus increased mineralization rates. More specific comments on this below.

In-stream processing. The conceptual model presented by the authors directly links stream data with riparian zone processes and thus ignore potential instream processing of the laterally-transferred DOC. Is this a relevant process in this catchment? More specific comments on this below.

Leaf litter directly falling into the stream. Leaf litter can be an important source of organic material including organic carbon in some forest headwater streams. In the aquatic compartment, this material can be dissolved, decomposed or just transported. Is this a relevant process in this catchment? More specific comments on this below.

The conceptual model would also benefit from some more literature cited to support some of the claims made.

I understand there are not data on groundwater tables, carbon pools or solute concentrations in the riparian zone available that could help to understand/support the mobilization process being proposed but maybe this could be mentioned and suggested for future studies.

A clearer description on what time periods were covered by the measurements for each of the variables presented in the study is needed. Particularly, it is not clear what period the weather variables cover. More specific comments on this below.

It would be nicer to see stream discharge presented in areally-normalized units (i.e. in mm) rather than in m3 s-1 so readers can relate to their catchments.

Specific comments

Title

The title is something very personal and chosen by the authors but what about "High-frequency measurements explain quantity and quality of dissolved organic carbon mobilization in a headwater catchment"?

Abstract

P. 1, L. 11-12. The exports are important but in the context of this sentence I think it is more accurate to mention concentrations. So please rephrase to "[. . .] (DOC) concentrations and exports from [. . .]" or simply to "[. . .] (DOC) concentrations from [. . .]".

P. 1, L. 14. It was a bit more than a one-year period actually, right?

P. 1, L. 20. Can you rephrase to "Selected statistical multiple linear regression models"?

P. 1, L. 25-27. Please, consider the comments I provide in relation to the interpretations provided in this sentence.

P. 1, L. 28. Which are or what type are these "few controlling variables"? Could you maybe rephrase to "few controlling hydroclimatic variables"?

Introduction

P. 2, L. 3. I am skeptical about the conclusions drawn by Freeman et al. (2001) and tend not to cite it.

P. 2, L. 19. Reduction in ionic strength is not by itself a cause of the increase in DOC concentrations but this mechanism is linked with the decline in atmospheric acid deposition that, in its turn, intensifies organic matter solubility by increasing humic charge and, indeed, reducing ionic strength. See e.g. Tipping and Hurley (1988). So please, remove that mechanism from the list or elaborate on the acid deposition process.

P. 2, L. 20-21. Please, either remove or briefly explain how median Ca and Mg represent sensitivity to acid deposition.

P. 2, L. 17-21. This paragraph is probably not critical but I like it and support the authors to keep it but I wonder if it could be merged somehow with the previous paragraph. It feels a bit out of place here.

P. 3, L. 3-6. In these context, see also the work done by Claire Tunaley in the Scottish highlands (e.g. Tunaley et al., 2016).

P. 3, L. 14. Quality and quantity dynamics?

P. 3, L. 17-18. Could you specify already here that the high-frequency measurements were done in a headwater stream? At this point it might still look like soil water measurements were done.

P. 3, L. 18. Could you write "the most decisive hydroclimatic factors"?

Materials and Methods

P. 3, L. 31. Do you mean "2.91 km km-2" instead of "2.91 km km-1"? I thought drainage density was given by unit of area.

P. 4, L. 1-7. This seems like a quite flat catchment with, consequently I would say, a large proportion of the total area covered by the riparian zone. Is this so? How does this headwater compared to similar headwaters in the temperate zone in this regard?

P. 4, L. 14. Strictly speaking, absorption spectroscopy is used to estimate dissolved organic matter quality, not just DOC quality, because absorbance reflect molecular structures of carbon and other elements. Please, mention this and maybe then you can say that for simplification and because carbon is the main focus of the paper you will talk about DOC quality.

P. 4, L. 17-18. You refer to origin first as either "autochthonous vs. allochthonous", which is fine but then you mention "molecular weights", which is not really an "origin" or does not directly refer to "origin" of the organic matter.

P. 4, L. 14-19. I think this paragraph describing the two optical parameters should be more elaborated. SUVA254 and S275-295 should be presented separately including for each of them: how they are calculated, what they mean, what one can infer from them, what the high vs. low values are and how they relate to with organic matter properties, and relevant references.

P. 4, L. 20. It was installed in April 2013, but when was it removed? How far does the time series go? It would be helpful to have more descriptions (and they should be consistent) of the periods of measurements for the different variables presented in the paper.

[Figure]

P. 4, L. 24-26. In the supplement S1 you mention that "before UV-Vis measurements were further processed". Maybe I am missing something but how many "before UV-Vis" measurements are in each case and how do you decide which measurements are classified as "before"?

P. 5, L. 23. Can you elaborate a bit more on how the events were "extracted"?

P. 5, L. 24-25. It would be helpful to know when the weather station started recording and for how long, i.e. the period of measurements. Because, does the weather time series actually cover the two months prior the beginning of the sensor measurements in the stream so that you can have e.g. AI60 values to work with? This point was not entirely clear to me and it is quite important to clarify.

P. 5, L. 24-25. How do the measurements from this weather station compared with the measurements from the weather station that was mention in the study site description?

P. 5, L. 30. Why did you chose 60 days as the reference to work with? I can see you also looked at AI6 and AI14 but there are many other options. Using AI60 seems a bit arbitrary.

P. 6, L. 2. Again, why 30 days?

P. 6, L. 4. "Analogous" instead of "complete"?

P. 6, L. 4-6. The description on how the different time periods for the different variable measurements overlap has to be made clearer.

P. 6, L. 18. Is this "n = 38" the number of events extracted with the method explain in P. 5, L. 23? Maybe mention it there then.

P. 6, L. 21. I am probably missing something but why is Qhf log and Qb is not log?

P. 6, L. 27. Please, write "hydroclimatic variables" instead of "environmental variables".

P. 7, L. 7. Please, remove "On the one hand".

[Figure]

Results

P. 7, L. 19-20. Maybe you can also add the average duration of these 38 discharge events, as well as indicate the average frequency in month-1 besides d-1.

P. 8, L. 8. "values less match the manual measurements" seems like an odd grammar construction.,

P. 8, L. 7-8. Define "extreme situations". This seems a bit vague.

P. 8, L. 4-10. I am a bit confuse here. I can see the PLS does a pretty good job on estimating DOC concentrations from the UV-Vis spectra and they resemble well the DOC concentrations measured in the lab, but then I don't understand why SUVA254 values measured at the lab are not as well captured. On the other hand, grab DOC does not really capture the whole range of DOC values, so that might be an issue. But if sensor and lab DOC values are very similar that can only mean that absorbance at 254 nm measured with the sensor significantly differ from that measured in the lab, right? Do you have any comparison of sensor versus lab 254 nm absorbance? Please, clarify this point.

P. 8, L. 31-32. Please, merged this sentence with the previous paragraph.

P. 9, L. 8. According to Table 2, Qb does not really correlate (high coefficient of determination) with CDOC or SUVA254.

P. 9, L. 11. If there are 38 events what is the average event duration to cover 47.5% of the entire time series? Seem like a lot.

P. 10, L. 13. Please, write "do" instead of "does".

P. 10, L. 25-26. Please, rephrase this sentence. There seem to be some verb missing.

P. 11, L. 28. Please, remove "rather".

Discussion
P. 12, L. 2-9. I would actually expect to see the "largest amounts of available DOC in the riparian zone" at the end of the summer or in early autumn (see e.g. Clark et al., 2005), basically at the end of the growing season, not necessarily in the summer or simply "when it is warm". Of course, actual DOC measurements in the riparian soil water would help to depict this and should be something to consider for the future.

P. 12, L. 22-24. As I can more or less guess from Figure 3, winter, spring, summer, and autumn are a bit sifted back, I guess because you are using antecedent conditions in your variables. Then I am not convinced about e.g. "cold and wet situations mainly found in winter" actually represent winter but likely also autumn. The same goes for all the other seasons.

P. 12, L. 31-32. Please, switch the order of the values for SUVA254 and S275-295 presented in these sentences so they are consistent with the order of presentation of the parameters.

P. 13, L. 3. Odd grammar, please rephrase.

P. 13, L. 25-27. The role of instream processing as well as of leaf litter falling directly into the stream (which can be a source of DOC) should be given more consideration as it might be important for the patterns you see in the stream so they might not directly connect to the riparian zone, or at least not as straight forward, especially when you do not have riparian zone measurements to back up your conclusions. It might be that the residence time of the water in the stream is too low to allow for instream processing to be important, or that leaf litter fall and subsequently leaf litter decomposition are not quantitatively important either, but if so, you need to argue it. This is a critical point to consider in your conceptualization.

P. 13, L. 29-31. When you talk about production and accumulation you cannot forget about mineralization. It might be that during dry and warm conditions the top soil is not hydrologically connected to the stream and thus that output of DOC from the system is non-existent, but precisely because you have those conditions you will have a larger

oxygen supply and increased mineralization rates (not only increase production). This is another way in which DOC can leave the system and would need to be compared with the production term in order to estimate net accumulation. You at least need to acknowledge this.

P. 14, L. 2. Higher SUVA254 values are commonly associated with higher aromaticity of the organic matter, rather than "processed", which might or might not be the case.

P. 14, L. 2-3. High SUVA254 values representing high aromaticity together with high S275-295 representing low molecular weight seems a bit conflicting.

P. 14, L. 6. There are better cites than Seibert et al. (2009) for the transmissivity feedback mechanism, e.g. Bishop et al. (2004) or, originally, Rodhe (1989).

P. 14, L. 6-10. Which would be these preferential flow paths? Lateral water transfer through unsaturated layers over the groundwater table? Do you expect to have this process in your catchment? Do you have any groundwater table measurements in the catchment that you can plot against stream discharge to understand this?

P. 14, L. 17. Odd grammar, please rephrase.

P. 14, L. 18-20. You probably have less production but you likely have less mineralization as well which need to be accounted when discussing net accumulation.

P. 14, L. 31-32. Please, rephrase this sentence, there seems to be a verb missing or the order of some words should be different.

P. 15, L. 5-6. But, in general, you have a positive relationship between DOC concentrations and stream discharge and that would not support limited availability of DOC in the riparian zone.

P. 15, L. 7-8. Impairs both production and mineralization.

P. 15, L. 9-10. Exactly, because of this hydrological connectivity with rich DOC sources I would not expect low DOC concentrations.

P. 15, L. 13. This has not been shown.

P. 15, L. 14. Do your soils freeze?

P. 15, L. 15. I would argue that depletion of exportable DOC sources due to low production is a bit too speculative as there is no information on soil and soluble pools in the riparian zone. And again, mineralization would be low as well.

P. 15, L. 24. Yes, I agree, the variance is low but that does not mean the absolute values are low.

P. 15, L. 28-29. The lack of whether data when? Was the period prior to the beginning of the sensor measurements not recorded for weather variables and so you could not use AI60 in your analyses after two months of sensor deployment? This needs to be clarify.

Conclusions

P. 16, L. 6-7. Again, mineralization is ignored here.

P. 16, L. 9-11. Exactly, wet situations are not mobilization limited and so they can lead to high DOC concentrations. And so I do not fully agree with the statement that high hydrological connectivity translate into low CDOC, because if the source is large and you seem to have a large riparian zone, this would not be the case.

P. 16, L. 17. This is the only place were decomposition is acknowledged as a process potentially occurring. This needs to be taking into consideration throughout.

P. 16, L. 23-27. Yes, and also actual measurements of riparian groundwater tables, riparian carbon pools and riparian soil water chemistry would be needed and helpful to understand this.

P. 16, L. 28-30. This sentence seems out of place.

P. 16, L. 31. "headwater" instead of "head water".

Figures and tables

Figure 2. Maybe you can leave the dates of the x-axis only in the lower panel and remove them from the other panels (as in Figure 3). Also, a different format of the dates (e.g. MM-YY) would allow for better visualization and more data points to be characterized. Specifically the beginning and end points of the axis should be labelled.

Figure 4. See previous comment on Figure 2 about the dates in the x-axis. Also, maybe thinner lines would improve visualization of the graphs.

Figure 6. My main problem with this figure is that in the warm & dry state you plot a higher CDOC in the soil but, again, what about the potentially high mineralization during this time. I would expect the highest CDOC concentrations at the end of the summer or at early autumn and specifically following warm and wet, not dry, periods.

Table 1. "statistical models" instead of "models". Also, it would be helpful to know what period those descriptive statistics are based on.

Table 2. All correlations are highly significant because of the large sample size. Worth mention it.

Table 3. "hydroclimatic variables" instead of "environmental variables".

Suggested references

Bishop, K., Seibert, J., Köhler, S., and Laudon, H.: Resolving the Double Paradox of rapidly mobilized old water with highly variable responses in runoff chemistry, Hydrological Processes, 18, 185-189, 10.1002/hyp.5209, 2004.

Clark, J. M., Chapman, P. J., Adamson, J. K., and Lane, S. N.: Influence of drought-induced acidification on the mobility of dissolved organic carbon in peat soils, Global Change Biology, 11, 791-809, 10.1111/j.1365-2486.2005.00937.x, 2005.

Rodhe, A.: On the generation of stream runoff in till soils, Nordic Hydrology, 20, 1-8, 1989.

Tipping, E., and Hurley, M. A.: A model of solid-solution interactions in acid organic soils, based on the complexation properties of humic substances, Journal of Soil Science, 39, 505-519, 1988.

Tunaley, C., Tetzlaff, D., Lessels, J., and Soulsby, C.: Linking high-frequency DOC dynamics to the age of connected water sources, Water Resources Research, 52, 5232-5247, 10.1002/2015wr018419, 2016.

---

## Referee Comment (RC2) · Anonymous Referee #2 · 9 Jun 2019

The manuscript describes a ∼ one year high frequency measurements of dissolved organic carbon (DOC) concentrations and quality in a river draining a headwater catchment in the Harz mountains, Germany. The authors measured DOC-concentrations and SUVA254 every 15 min by means of a UV-Vis probe and calculated the spectral between 275 and 295 nm as indicator of DOM quality. Validation and calibration of both parameters were performed through SAC254 and DOC measurements of grab samples. Discharge was calculated from a stage-discharge relationship based on data obtained from pressure transducer and manual discharge measurements. From their DOC measurements the authors concluded that changes in DOC concentrations and quality are mainly determined by antecedent hydroclimatic conditions. Due to this,

the authors used an aridity index (AI60) as well as temperature and discharge of the preceeding 30 days and a so called discharge normalized temperature index as controlling variables/predictors to statistically model DOC concentrations changes in their river. From their data they suggested a conceptual model which differentiates between three hydroclimatic states (warm and dry, cold and wet and intermediate. The authors concluded that the DOC concentration variability in their stream can be predicted by a small number of controlling variable which are linked to DOC source activation, discharge events and seasonal changes in the DOC production in the riperian zone. "High frequency" measurements of DOC release from headwater catchments have been carried out in a number of studies before which showed broadly the same results and conclusions. For example, Broder and Biester, 2015 (also BGC) and Birkel et al 2017 published a study of high frequency measurements of DOC release from a peatland and forest catchment in the Harz (just a few kilometers away) and also modeled DOC release dependent on antecedent moisture conditions. Unfortunately, these papers have not been cited or discussed in the manuscript. What is new in this study is the really high frequency DOC monitoring (15 min) and the different statistically approaches. However, for me it is not really clear what actually the aim of the paper is. One reason for this might be that the paper lacks a clear hypothesis. Even the title does not contain a research question, just a statement of what has been done. The description of the aim of the study (p3) is quite general. . ... to obtain a better understanding. . .Looking at the conclusions, I don't think that the paper really provides more understanding than what is already known. It seems, that the authors cannot really decide if this is a eco-hydrochemical or a statistical-hydrological study. The value of the presented findings is difficult to evaluate as the authors have largely missed to compare and discuss their results to/with those of other studies. From what the authors stated in their (long) conclusions. . ..."Yet, it remains unclear which explicit mechanisms in the riparian zone are responsible for the measured and conceptualized DOC dynamics in the Rappbode stream. . ...... Further research is necessary to identify the explicit spatio-temporal mobilization patterns as well as molecular markers that can be used to trace DOC from

riparian source zones into the stream in order to fully understand DOC mobilization in the riparian zone."I think that is where other studies have ended before. The biogeochemical findings in this study are quite limited, so that the study has its emphasis on the statistical approach which is clearly necessary to extract a message from the large (high frequency) data set. However, as the authors base their predictors on 60 and 30 means, the meaning of the high-frequency DOC monitoring remains form e unclear. I think it would be interesting to use this data set to evaluate which frequency is at least necessary to capture the role of the predictors and the magnitude of DOC concentration/flux changes (38 discharge events!). Moreover, there are several factors in this data set which might be interesting to evaluate regarding the sensitivity of the model towards the predictors e.g. the magnitude of DOC-flux changes during discharge events, the role of catchment size, DOC-pools etc. but are not discussed.

This manuscript is in general suitable for publication in BGC. I also think that the quality of the data and the approach is good. However, I think before this manuscript can be accepted the authors should try to give their manuscript a clearer aim/hypothesis which goes beyond a gererally better understanding of what is already known. I suggest, that the authors extend their introduction by other studies (there are numerous) on this topic. From this they can probably better deviate what is already known and what the (new) aim of their study is (why needs the frequence be higher than in other studies ?). Similar, they should extend their discussion with a comparison to data from other studies and the sensitivity and potential limitations of their predictors including the characteristics of the catchment and a discussion on high frequent high frequency should be.

References:

Broder, T., Biester, H. Hydrologic controls on DOC, As and Pb export from a polluted peatland - the importance of heavy rain events, antecedent moisture conditions and hydrological connectivity. Biogeosciences, 12, 4651-4664 (2015).

[Figure]

Birkel, C., Broder, T., Biester, H. (2017). Nonlinear and threshold-dominated runoff generation controls DOC export in a small peat catchment. Journal of Geophysical Research: Biogeosciences, 122, 498-513.
* * *

---

## Referee Comment (RC3) · Anonymous Referee #3 · 11 Jun 2019

General Comments: In this paper entitled "High-frequency measurements of dissolved organic carbon quantity and quality in a headwater catchment" Benedikt et al. report the results of a high-frequency monitoring study in a small, alpine catchment in Germany. Using 15-minute interval data, the authors analyzed correlations between temporal variation in dissolved organic carbon (DOC) quantity and optical properties on event and seasonal scales with antecedent weather conditions. They found that seasonal state influenced DOC concentration as well as the response to change in discharge. This work is a strong, observational study that build off the growing literature on high-frequency DOC dynamics. The paper is well written and citations are generally appropriate, though there are gaps, which is inevitable in large field such as

this. I was also impressed by the clarify and creativity of the figures, which can be challenging for an observational paper with so much data. The combination of high-quality and unique data with a thoughtful statistical framework makes this a compelling and complex study. With a thorough revision, this paper would be a strong contribution to Biogeosciences. A few general comments and suggestions below:

1. There is a clear goal stated at the end of the introduction, but there are not clear hypotheses until the conceptual model is presented (Figure 6). I think that putting the conceptual framework at the front of the paper (introduction or at latest the methods) would help the reader understand how the authors are viewing the system, better appreciate the findings, and better grasp why certain methods were used. 2. The paper spends quite a bit of time discussing long-term trends in DOC attributed to changes in acid deposition, land use, and climate change. This focus was something of a red herring, as the paper is strongest on a much shorter timescale, which does not speak directly to this literature. Additionally, most of the cited papers on DOC trends are older, which I think is a recognition that while many regional trends exist (for either increasing or decreasing DOC), there is not a clear pattern or signal of anthropogenic effects on DOC concentration. There is more evidence of anthropogenic effects on DOC properties (e.g. Butman et al., 2014), and this could be fruitful, but, I think the ecohydrological focus on sources and fate of DOC is most compelling. This fits in better with the conceptual model and approach of the paper. There are many other reasons to study DOC, many of which are brought up elsewhere in the introduction (Zarnetske et al., 2018), so starting the paper with this observation is less effective. 3. The discussion seemed somewhat uneven to me—with the authors still defining some concepts and findings and even describing methods. I think that reorganizing the paper around a clear set of hypotheses would strengthen this already interesting piece of work. 4. Throughout the paper, I was surprised at the lack of discussion of interactions with other elements. DOC does not cycle in isolation, and stoichiometry can have a strong influence on DOC production and consumption (Helton et al., 2015). not to find greater discussion of DOC removal mechanisms, including heterotrophic respiration and abiotic removal (Raymond et al., 2016). I imagine that nitrogen and phosphorus data are available (NO3 data, specifically should be available through the whole time period), and including and integrating them could greatly strengthen the paper. For example, how do N and P vary during the chosen seasonal periods and how might that influence temporal patterns currently attributed to changes in source and transport limitation?

References Butman DE, Wilson HF, Barnes RT, Xenopoulos MA, Raymond PA. 2014. Increased mobilization of aged carbon to rivers by human disturbance. Nature Geoscience 8 (2): 112–116 DOI: 10.1038/ngeo2322 Helton AM, Ardón M, Bernhardt ES. 2015. Thermodynamic constraints on the utility of ecological stoichiometry for explaining global biogeochemical patterns (P Jeyasingh, ed.). Ecology Letters 18 (10): 1049–1056 DOI: 10.1111/ele.12487 Raymond PA, Saiers JE, Sobczak WV. 2016. Hydrological and biogeochemical controls on watershed dissolved organic matter transport: pulse-shunt concept. Ecology 97 (1): 5–16 DOI: 10.1890/14-1684.1 Zarnetske JP, Bouda M, Abbott BW, Saiers J, Raymond PA. 2018. Generality of Hydrologic Transport Limitation of Watershed Organic Carbon Flux Across Ecoregions of the United States. Geophysical Research Letters 45 (21): 11,702-11,711 DOI: 10.1029/2018GL080005

---

## Referee Comment (RC4) · Anonymous Referee #4 · 16 Jun 2019

General Comments: The authors report on a study of DOC concentration and quality over time in a stream using a combination of grab samples and an in situ UV-Vis spectrophotometer. The authors find that during warm, dry periods DOC concentrations were highly responsive to storm events (increases in discharge). As conditions became wetter and cooler, the responsiveness of DOC concentrations decreased. The authors attribute this to differences in hydrologic flowpaths and DOC mobility and production across landscapes. The authors use a thorough analysis to support their arguments and conclude the paper with an interesting conceptual framework that synthesizes their own findings with other existing literature. I think that the authors present a strong study, but have some suggestions for clarity and organization

[Figure]

I argue that the authors have over-emphasized the trend of increasing DOC flux trends in the abstract and introduction. This is an important reason to study this subject and this work could certainly be used to better understand the mechanisms driving this trend, but the paper includes neither a report on this increasing trend or evidence for a mechanism for this trend. I think that most of the parts of the introduction are there, but I suggest that the text focus more on the aspects of DOC export reported on in the paper (transport of DOC from watersheds across hydrologic regimes and antecedent conditions). One thing that is missing from the introduction is any mention of antecedent moisture conditions. I think that discussing the role of antecedent conditions, discharge-normalized temperature, and their potential role on DOC quantity and quality should be discussed before the methods section since these are a major focus of the paper and the conceptual model discussed later (figure 6).

Specific Comments: Page 3; Lines 14-16: I highlight this sentence because I think that it does an excellent job of encapsulating this study. I suggest the authors reorganize the introduction to better emphasize concepts related to this idea.

Page 4; Line 27: Were grab samples also run for spectral slope values in addition to UVA-254? I also suggest that the authors provide some additional detail about sample collection (e.g., filter size, sample handling).

Page 4; Line 29: Some brief information about methods for DOC analysis (e.g. acidification level) would be of use for the reader.

Page 5; Line 6: I suggest the authors report how closely SUVA values obtained from the sensor match the grab sample values.

Page 6; Line 21: I'm concerned that hysteresis loop size is biasing regression slopes obtained from this method. I would appreciate some support for application of this method to events with varying degrees of hysteresis.

Page 7; Line 27: Reporting an actual mean AI or similar number would be helpful for

the reader.

Page 8; Line 28: I think that there are better ways to present this information. As written, it is hard for the reader to tell what the readers should take away from this paragraph.

Page 9; Line 16: Presenting this information here is repeatitive. I argue this authors should move some of this material to the methods section where similar methods are already covered.

Page 9; Line 30: I am concerned about the interpretation of individual regression coefficients from a multiple regression of observational data due to issues of multicollinearity. In other parts of the analysis, partial least squares regression is used to address this issue, but for this analysis it appears that multiple linear regression was used instead (Page 6, Line 18).

Page 10; Lines 9-20 and Table 3: Values of a change dramatically depending on whether a 15 or 30 day lag is used. For example, a (CDOC) is negatively correlated to T15, but positively correlated to T30. The same is true for Q15 vs Q30. This pattern is reportead for a (SUVA254) and a (S275-295). It would be interesting to see if the correlation between a and DNT15 is negative. This would seem to change the implications of the study substantially.

Page 10; Line 28: Referring to this analysis as seasonal-scale is somewhat confusing to me because there is no specific season analysis conducted here (rather variables that AI that typically change with season are used). I would also like to know if any data were held out for model validation or if the R2 statistics in Table 4 are for the model with reference to the training dataset alone.

Page 13; Lines 9-12: The different weather scenarios make sense, but I think that stating that there are three discrete states is a bit arbitrary and is not supported by any sort of data or analysis. I think the general framework is right and it makes sense

[Figure]

for the authors to highlight certain scenarios. That said, I don't see evidence in the data for discrete states but for a continuum without jumps from one state to another. I recommend the authors clarify the nature of their conceptual model.

Figure 6: How baseflow DOC concentration changes with season was not supported with particular numbers in the results, but appears to important for the conceptual framework. It is discussed briefly in a qualitative fashion, but the degree of seasonal differences in DOC concentrations are hard for the reader to infer.

———————————————————

---

## Author Comment (AC1) · 1 Aug 2019

Dear Referee #1,

We are very grateful for the detailed and valuable comments on our manuscript. We are confident that we can address most of the comments satisfactorily. Special emphasis will be put on a proper biogeochemical discussion of mineralization and in-stream processing to better demarcate DOC net production. This will greatly strengthen our proposed conceptual model. Our point-by point response to the comments is available by the link below, with the reviewer's comments (black) quoted first and followed by our response (red).

[Figure]

On behalf of all co-authors,

Benedikt Werner

Please also note the supplement to this comment:
https://www.biogeosciences-discuss.net/bg-2019-188/bg-2019-188-AC1-supplement.pdf

**Supplement:**

Response to **Anonymous Referee #1**

General comments
My main concern is that, in my opinion, some important processes for the context of
the present study, and particularly for the conceptual model proposed, are ignored.
Specifically I refer to:
Mineralization (decomposition/respiration). When discussing DOC accumulation in the
riparian zone, besides talking about production versus lateral mobilization one needs
to account for mineralization, which is another way in which DOC can be lost. For
example, the authors claim that warm and dry conditions are optimal for DOC accumulation
because of increased production rates and low hydrological connectivity but
these situations can also favour high oxygen supply and thus increased mineralization
rates. More specific comments on this below.

(R1GC1)
We appreciate your evaluation of our Manuscript (MS). We realized that a proper discussion of
biogeochemical processes was not clearly enough addressed.

The reviewer is correct in asserting that lower water contents can increase the mineralization rate
compared to water-logged soils. However, literature data (Boissier and Fontvieille, 1993; Boyer and
Groffman, 1996; Grøn et al., 1992; Nelson et al., 1994; Yano et al., 1998) show that 56% or more of
the DOC in the soil solution of forest soils is poorly biodegradable. This portion of the accumulated
carbon will presumably still be available for transport towards streams even if mineralization rates
increase. Furthermore, in carbon-rich top soils mineralization and DOC accumulation do not appear
to have an either-or attribute: Kalbitz et al. (2000) and citations therein report a positive correlation
between mineralization rate and DOC concentration of the soil solution.
Given the nature of our monitoring approach and the research questions we were addressing by it,
the paper focuses on the hydrolclimatological drivers of DOC transport towards streams. While this
approach finds support by Kalbitz et al.'s (2000) conclusion that hydrology dominates over biology in
determining DOM fluxes, we also see the validity of the revier's concern in this regard.
In consequence one can state that DOC production can be higher than mineralization in the shallow
water table environment of riparian zones (Ledesma et al. 2018) leading to a **net production** of DOC.
We will therefore clarify the focus of the paper and add a discussion of the role of mineralization that
will address the various comments on the topic by the reviewer. The term "net production" will be
carefully defined and used throughout the MS to avoid ambiguities.

In-stream processing. The conceptual model presented by the authors directly links stream data with riparian zone processes and thus ignore potential instream processing of the laterally-transferred DOC. Is this a relevant process in this catchment? More specific comments on this below.
+
Leaf litter directly falling into the stream. Leaf litter can be an important source of organic material including organic carbon in some forest headwater streams. In the aquatic compartment, this material can be dissolved, decomposed or just transported. Is this a relevant process in this catchment? More specific comments on this below.

(R1GC2)

Without doubt, there will be to some extend instream processing occurring and leaf litter leaching in this catchment. However, there are several reasons speaking for a minor impact of instream processes up to our study site:

1 - first of all, routine measurements at our study site (mostly during non-event situations) showed a BIX consistently below 0.7 indicating allochthonous dominated waters (Huguet et al., 2009). This is in line with Creed et al. (2015),Nimick et al. (2011) and citations therein, stating that in general headwaters are dominated by allochthonous carbon with the role of instream processing increasing with stream order. The role of instream processing during event flows furthermore should decrease in comparison to that of low flow situations due to hydrodynamic scaling (a shorter residence time and relatively less hyporheic exchange of stream water). Also strong increasing DOC concentrations during events which could further mask the impact of instream processing.

2 - Köhler et al. (2002) showed that within short time scales (<1d) changes from DOC processing (degradation and photobleaching) in incubation experiments were minimal in a headwater catchment in Sweden. Even during baseflow situations, average hydrological residence time in the Rappbode catchment should be below one day (roughly 2km downstream from the spring) and thus a relatively small exposure/reaction time with regard to instream processing has to be expected. Note that the wide riparian zone (several tens of meters) in our catchment consists to large parts of a flood plain, leaving only little possibility for leaf litter falling directly into the stream. As stated above, residence time scales in the studied stream are rather low which further impedes significant leaf litter leaching (which occurs in timescales of around 24h (Dowell, 1985; Kaplan et al., 2008).

For clarification we will change the MS by elaborating on the importance of instream processing with respect to our catchment setting (see also specific comments below).

The conceptual model would also benefit from some more literature cited to support some of the claims made.
(R1GC3)
We agree. Additional supporting claims of the conceptual model (e.g. support for seasonal and temperature controlled changes in soil DOC concentration (Kalbitz et al., 2000) and citations therein) will be included in the MS where appropriate.

I understand there are not data on groundwater tables, carbon pools or solute concentrations in the riparian zone available that could help to understand/support the mobilization process being proposed but maybe this could be mentioned and suggested for future studies.
(R1GC4)

We agree with the reviewer and will mention this in the revised MS (see also RC31). But we also want to stress that the paper demonstrates the considerable value of high-frequency measurements of DOC quality and quantity in unraveling DOC mobilization in the riparian zone without the need for additional data collection beyond the stream. We believe this allows long-term DOC monitoring with a manageable allocation of time and resources.

A clearer description on what time periods were covered by the measurements for each of the variables presented in the study is needed. Particularly, it is not clear what period the weather variables cover. More specific comments on this below.
(R1GC5)
We agree. A more detailed description of the coverage of the measurements will be incorporated in the MS (see specific comments on this below).

It would be nicer to see stream discharge presented in areally-normalized units (i.e. in mm) rather than in m³ s⁻¹ so readers can relate to their catchments.
(R1GC6)
We agree partially. The units of choice give a good impression of the size of the stream, which will also be useful to the readership. To facilitate comparison to differently-sized catchments, we will add an overview of the specific discharge in the section with descriptive statistics to address this comment.

Title

The title is something very personal and chosen by the authors but what about "Highfrequency measurements explain quantity and quality of dissolved organic carbon mobilization in a headwater catchment"?
(R1C1) (Referee#1 Comment 1)
This is an interesting suggestion. We will incorporate this in the revision.

Abstract
P. 1, L. 11-12. The exports are important but in the context of this sentence I think it is more accurate to mention concentrations. So please rephrase to "[: : :] (DOC) concentrations and exports from [: : :]" or simply to "[: : :] (DOC) concentrations from [: : :]".
(R1C 2) We agree. The sentence will be changed to "[: : :] (DOC) concentrations and export from [: : :]"
P. 1, L. 14. It was a bit more than a one-year period actually, right?
(R1C 3) Yes. The sentence will be changed to "A roughly one-year high-frequency (15 minutes) dataset [..]".
P. 1, L. 20. Can you rephrase to "Selected statistical multiple linear regression models"?
(R1C 4) Changes will be incorporated as proposed.
P. 1, L. 25-27. Please, consider the comments I provide in relation to the interpretations provided in this sentence.
This will be taken into consideration (see also R1GC1).

P. 1, L. 28. Which are or what type are these "few controlling variables"? Could you maybe rephrase to "few controlling hydroclimatic variables"?
(R1C5) Changes will be incorporated as proposed.

Introduction
P. 2, L. 3. I am skeptical about the conclusions drawn by Freeman et al. (2001) and tend not to cite it.
(R1C6) The citation of Freeman et al. (2001) will be removed.

P. 2, L. 19. Reduction in ionic strength is not by itself a cause of the increase in DOC concentrations but this mechanism is linked with the decline in atmospheric acid deposition that, in its turn, intensifies organic matter solubility by increasing humic charge and, indeed, reducing ionic strength. See e.g. Tipping and Hurley (1988). So please, remove that mechanism from the list or elaborate on the acid deposition process.
+
P. 2, L. 20-21. Please, either remove or briefly explain how median Ca and Mg represent sensitivity to acid deposition.
+
P. 2, L. 17-21. This paragraph is probably not critical but I like it and support the authors to keep it but I wonder if it could be merged somehow with the previous paragraph. It feels a bit out of place here.
(R1C7) The paragraph will be deleted (see general comments on the introduction in answer to Referee #3 R3GC1 and Referee#4 R4GC1).

P. 3, L. 3-6. In these context, see also the work done by Claire Tunaley in the Scottish highlands (e.g. Tunaley et al., 2016).
(R1C8) Tunaley et al., 2016 fits well to the context and will be incorporated in the MS as followed:
"Recent years have seen significant advances in sensing technologies for high-frequency in situ concentration measurements (Rode et al., 2016; Strohmeier et al., 2013), facilitating the assessment of DOC delivery to streams (Tunaley et al., 2016). "

P. 3, L. 14. Quality and quantity dynamics?
+
P. 3, L. 17-18. Could you specify already here that the high-frequency measurements were done in a headwater stream? At this point it might still look like soil water measurements were done.
+
P. 3, L. 18. Could you write "the most decisive hydroclimatic factors"?
(R1C9) We agree. All suggestions will be incorporated in the revised text.

Materials and Methods
P. 3, L. 31. Do you mean "2.91 km km$^{-2}$" instead of "2.91 km km$^{-1}$"? I thought drainage density was given by unit of area.
(R1C10) We agree. We apologize for the mistake. This will be changed in the MS.

P. 4, L. 1-7. This seems like a quite flat catchment with, consequently I would say, a large proportion of the total area covered by the riparian zone. Is this so? How does this headwater compared to similar headwaters in the temperate zone in this regard?
The catchment is in a hilly region. The stream is flanked by a riparian zone with a slope towards the stream bed that is small compared to the slope in the main direction of the stream. This relatively flat riparian zone lies between much steeper forested slopes. This topography leads to a riparian zone that is wet most of time, and thus offers conditions favorable to DOC export to its stream. The catchment's 90$^{th}$ percentile of the topographic wetness index, standing for the abundance of riparian wetlands (Musolff et al., 2018) is 8.3, quite close to the TWI-90$^{th}$ median of 89 catchments across Germany presented in Musolff et al. 2018). The same is true for the land use (median of 79% in the 89 German catchments). We can therefore state that the study catchment is representative in

topography and land use for an average catchment in Germany draining to a drinking water reservoir. We will add that information to the text at this point.

P. 4, L. 14. Strictly speaking, absorption spectroscopy is used to estimate dissolved organic matter quality, not just DOC quality, because absorbance reflect molecular structures of carbon and other elements. Please, mention this and maybe then you can say that for simplification and because carbon is the main focus of the paper you will talk about DOC quality.
+
P. 4, L. 17-18. You refer to origin first as either "autochthonous vs. allochthonous", which is fine but then you mention "molecular weights", which is not really an "origin" or does not directly refer to "origin" of the organic matter.
+
P. 4, L. 14-19. I think this paragraph describing the two optical parameters should be more elaborated. $SUVA_{254}$ and $S_{275-295}$ should be presented separately including for each of them: how they are calculated, what they mean, what one can infer from them, what the high vs. low values are and how they relate to with organic matter properties, and relevant references.

(R1C11) The paragraph will be rewritten; the proposed changes will be addressed by: "We used in situ absorption spectroscopy to estimate dissolved organic matter quantity and quality. For simplification and because carbon is the main focus of the paper, dissolved organic matter quality will be addressed as DOC quality in the following. DOC quality can be characterized by specific metrics based on the light absorbing properties of dissolved organic compounds: $SUVA_{254}$ [L m$^{-1}$ mg-C$^{-1}$] is the spectral absorption coefficient at 254 nm (SAC254) [m$^{-1}$] divided by the $C_{DOC}$ [mg L$^{1-}$]. $SUVA_{254}$ correlates well with aromaticity of DOC and therefore can be used as an indicator of the general chemical composition and reactivity of organic carbon (Weishaar et al., 2003). To refine the understanding of DOC composition, the spectral slope between 275 and 295 nm, denoted $S_{275-295}$ was estimated from the adsorption spectra and calculated as described in Helms et al. (2008): A linear regression model was fitted for each time step to the logarithms of the absorption coefficients between 275 and 295 nm to derive the slope $S_{275-295}$. $S_{275-295}$ can help to distinguish between autochthonous and allochthonous DOC, molecular weights and processing (photobleaching and microbial degradation change aromaticity) (Helms et al., 2008). The general patterns of such DOC quality metrics can be used to infer information on origin and properties of DOC and thus to characterize source zones of DOC in riparian zones (Eran et al., 2006; Hutchins et al., 2017; Sanderman et al., 2009). "

P. 4, L. 20. It was installed in April 2013, but when was it removed? How far does the time series go? It would be helpful to have more descriptions (and they should be consistent) of the periods of measurements for the different variables presented in the paper.

(R1C12) The data set ends in October 2014. This will be indicated in the revised version.

P. 4, L. 24-26. In the supplement S1 you mention that "before UV-Vis measurements were further processed". Maybe I am missing something but how many "before UVVis" measurements are in each case and how do you decide which measurements are classified as "before"?

(R1C13 and R1CS1, resp.) Indeed, this is written a little bit confusing. For clarification, the sentence will be changed to: "Ahead of further (statistical) processing, each of the UV-Vis absorption spectra was corrected for this drift by subtracting an exponential function fitted to the raw data."

P. 5, L. 23. Can you elaborate a bit more on how the events were "extracted"?

(R1C14) The MS will be changed and an elaboration of the event extractions will be included as follows: "Consequently, subtracting the baseflow hydrograph ($Q_b$) from the total hydrograph of $Q_{tot}$

yields the hydrograph of $Q_{hf}$, which has positive values during events ($Q_{tot} > Q_b$) and is zero during non-event periods (when $Q_{tot} = Q_b$). All continuous positive values between two non-event periods (zero values) were considered as one event and extracted from the complete dataset for further processing."

P. 5, L. 24-25. It would be helpful to know when the weather station started recording and for how long, i.e. the period of measurements. Because, does the weather time series actually cover the two months prior the beginning of the sensor measurements in the stream so that you can have e.g. $AI_{60}$ values to work with? This point was not entirely clear to me and it is quite important to clarify.
+
P. 5, L. 24-25. How do the measurements from this weather station compared with the measurements from the weather station that was mention in the study site description?
(R1C15)
1) The weather station was activated in May 2013 after the various sensors were installed. Hence, to obtain a complete dataset of all measured parameters and its derivatives, modeling of DOC had to start 60 days later, at the end of July.
2) Comparison between the two weather stations showed a good agreement between both stations ($r_{spearman}$ = 0.7) yet there exist events  that were only captured by one of the weather stations. Changes will be made accordingly: "Measurements of the weather station started at 21 May 2013 until 26 November 2014. Measurements were at an hourly interval for the first five days, until 26 June 2013. […] As a consequence, time series of lumped variables start t days after the actual begin of the field observations."

P. 5, L. 30. Why did you chose 60 days as the reference to work with? I can see you also looked at $AI_6$ and $AI_{14}$ but there are many other options. Using $AI_{60}$ seems a bit arbitrary.
+
P. 6, L. 2. Again, why 30 days?
(R1C16) We chose $AI_{60}$ and $DNT_{30}$ as these variables turned out to work best in terms of variance inflation and interaction for the statistical modeling. This will be indicated in the revised version.
P. 6, L. 4. "Analogous" instead of "complete"?
+
P. 6, L. 4-6. The description on how the different time periods for the different variable measurements overlap has to be made clearer.
(R1C17) We agree. Changes will be implemented as follows: "In order to obtain an analogous dataset, time series of all variables were constrained by excluding such observations that fell into the data gaps of the UV-Vis probe (R1Cf. 2.2.1)."

P. 6, L. 18. Is this "n = 38" the number of events extracted with the method explain in P. 5, L. 23? Maybe mention it there then.
(R1C18) Yes. This will be changed in the MS to "(n = 38, extracted with the method explained in 2.2.2)".

P. 6, L. 21. I am probably missing something but why is $Q_{hf}$ log and $Q_b$ is not log?
(R1C19) In P.6, L. 15-20 we state "According C-Q and quality-Q relationships […] were represented by combinations of multiple linear regression models with $Q_{tot}$, $Q_b$ and $Q_{hf}$ and their log transformations as predictors. The best overall combination […] was chosen according to the best mean $R^2$ […]".

P. 6, L. 27. Please, write "hydroclimatic variables" instead of "environmental variables".
(R1C20) This will be changed accordingly in the MS.

P. 7, L. 7. Please, remove "On the one hand".
This will be removed in the MS.

Results
P. 7, L. 19-20. Maybe you can also add the average duration of these 38 discharge events, as well as indicate the average frequency in month$^{-1}$ besides d$^{-1}$.
(R1C21) The MS will be changed to: "[…] yielding an average frequency of 0.086 d$^{-1}$ (2.58 month$^{-1}$) at an average duration of 134 h per discharge event."

P. 8, L. 8. "values less match the manual measurements" seems like an odd grammar construction.
+
P. 8, L. 7-8. Define "extreme situations". This seems a bit vague.
(R1C22) We agree. The MS will be changed to: "[…] due to extreme situations such as droughts and floods which can strongly differ in DOC source area mobilization in comparison to average events (Vaughan et al., 2017). Accordingly, $C_{DOC}$ and hence calculated SUVA$_{254}$ values match the manual measurements to a lesser extent during such situations, […]"

+
P. 8, L. 4-10. I am a bit confuse here. I can see the PLS does a pretty good job on estimating DOC concentrations from the UV-Vis spectra and they resemble well the DOC concentrations measured in the lab, but then I don't understand why SUVA$_{254}$ values measured at the lab are not as well captured. On the other hand, grab DOC does not really capture the whole range of DOC values, so that might be an issue. But if sensor and lab DOC values are very similar that can only mean that absorbance at 254 nm measured with the sensor significantly differ from that measured in the lab, right? Do you have any comparison of sensor versus lab 254 nm absorbance? Please, clarify this point.

SUVA values were derived from field measurement of SAC254 with a handheld device and DOC measurements in the lab. They were only taken as a validation, but not calibration.
Both SAC254 and DOC values derived from the UV-Vis probe fit well to the field/ laboratory values: R² of SAC254 of the probe and handheld field values was 0.94 and R² of DOC fitted by PLSR was 0.97. However, SUVA is calculated as the ratio of SAC254 and $C_{DOC}$. The smaller the $C_{DOC}$ gets (these values were also in the calibration range!), the more sensitive SUVA values will be on systematic errors of the lab measurement and inaccuracies of the method (e.g. small deviation of the timing in in-situ values and grab sample taken). This was also shown in the MS: by removing three values which were measured during more extreme situations with low $C_{DOC}$, the R² of the fit increased from 0.5 to 0.73.Ggiven the good fit between SAC254 and DOC values of UV-Vis and lab measurements and the fact that SAC254 and DOC were derived from the same UV-VIS probe, we think that also UV-Vis derived SUVA$_{254}$ values should be reliable and consistent.

P. 8, L. 31-32. Please, merged this sentence with the previous paragraph.
This will be changed in the MS.

P. 9, L. 8. According to Table 2, $Q_b$ does not really correlate (high coefficient of determination) with $C_{DOC}$ or SUVA$_{254}$.
+
P. 9, L. 11. If there are 38 events what is the average event duration to cover 47.5% of the entire time series? Seem like a lot.

(R1C23)

P. 9, L. 8: We agree. It will be clarified in the MS that $Q_b$ only correlates with $S_{275-295}$.

P. 9, L. 11: We agree. This number is wrong. Events cover 44% of the entire time series. Calculation was conducted as follows:

Average duration of discharge events was 134 h (see C21). For 38 discharge events this results in ~222 days of discharge events for the entire time series.

The entire time series covers the period of 21 May 2013 until 08 October 2014 (~505 days). The ratio between 222 and 505 equals 0.44. Please consider that also snowmelt was incorporated as well as the fact that the recession curves of discharge events can last quite longer as the actual precipitation event. We apologize for the mistake and will change the event duration to 44% of the entire time series.

P. 10, L. 13. Please, write "do" instead of "does".
This will be changed in the MS.

P. 10, L. 25-26. Please, rephrase this sentence. There seem to be some verb missing.
(R1C24) The MS will be changed to: "We added to the model of Eq. (2) the seasonal-scale $AI_{60}$ and $DNT_{30}$. In addition we added those interactions for which VIF < 10 (Eq. (1)): $\log(Q_{hf})xQ_b$, $AI_{60}xDNT_{30}$ and $DNT_{30}xQ_b$. These two additions allow the model to account for temporal changes in the relationships of $C_{DOC}$ and DOC quality with discharge."

P. 11, L. 28. Please, remove "rather".
This will be changed in the MS.

Discussion

P. 12, L. 2-9. I would actually expect to see the "largest amounts of available DOC in the riparian zone" at the end of the summer or in early autumn (see e.g. Clark et al., 2005), basically at the end of the growing season, not necessarily in the summer or simply "when it is warm". Of course, actual DOC measurements in the riparian soil water would help to depict this and should be something to consider for the future.
(R1C25) We agree to see the largest amounts of available DOC in the riparian zone in end of summer/early autumn. We further agree that the term "when it is warm" is not suitable for the MS. We also agree that actual DOC measurements in the riparian soil would help to elucidate when there are the "largest amounts of available DOC in the riparian zone".
We will change this in the MS as follows: "The regression models for the discharge events revealed that discharge had a seasonally differing impact on DOC concentration and quality observed in the Rappbode stream (Fig. 3). Although the largest amounts of exportable DOC are to be expected at the end of the summer and in early autumn (Clark et al., 2005), $C_{DOC}$ and DOC quality changed most distinctly with the discharge components $Q_{hf}$ and $Q_b$ in the summer (Fig. 3).There were no DOC measurements of the riparian soil water available which could further elucidate this discrepancy. We argue that during summer initial $C_{DOC}$ was low during baseflow while large amounts of DOC were already available to be transported from the riparian soils to the stream during events. Overall this could explain the steep model coefficients $a$ in summer."

P. 12, L. 22-24. As I can more or less guess from Figure 3, winter, spring, summer, and autumn are a bit sifted back, I guess because you are using antecedent conditions in your variables. Then I am not convinced about e.g. "cold and wet situations mainly found in winter" actually represent winter but likely also autumn. The same goes for all the other seasons.
(R1C26) We agree that "cold and wet situations" could also represent (late) autumn, which is why we wrote "mainly found in winter". For Figure 3 we took the meteorological begin of the seasons (01

March instead of 21 March for the beginning of spring and so on) which might give an additional impression of back shifting (Figure 3 will be changed accordingly for better readability). Also seasonality relates to long time observations which can shift more or less strongly from year to year. Hence seasonal transition times (like late autumn) will fall into the "mainly winter" time for some years but not for others. Therefore situations were chosen in such a way that we could avoid potentially ambiguous seasonal terminology.

P. 12, L. 31-32. Please, switch the order of the values for SUVA$_{254}$ and S$_{275-295}$ presented in these sentences so they are consistent with the order of presentation of the parameters.
+
P. 13, L. 3. Odd grammar, please rephrase.

(R1C27) We agree, the DOC quality metrics will be reordered and P.13, L.3 will be changed to: "Daily mean C$_{DOC}$, SUVA$_{254}$ and S$_{275-295}$ values of 1.49 mg L$^{-1}$, 0.68 L m$^{-1}$ mg-C$^{-1}$ and 5 x10-3 nm$^{-1}$ were minimal at the end of the drought in August 2013, when baseflow levels were low, whereas values of 4.14 mg L$^{-1}$, 4.05 L m$^{-1}$ mg-C1 -and 15.8 x10-3 nm$^{-1}$ were(?) measured […]. Events during the intermediate state also showed elevated C$_{DOC}$, SUVA$_{254}$ and S$_{275-295}$ values, but in comparison to summer events with a decreased variance and range (Fig. 5)."

P. 13, L. 25-27. The role of instream processing as well as of leaf litter falling directly into the stream (which can be a source of DOC) should be given more consideration as it might be important for the patterns you see in the stream so they might not directly connect to the riparian zone, or at least not as straight forward, especially when you do not have riparian zone measurements to back up your conclusions. It might be that the residence time of the water in the stream is too low to allow for instream processing to be important, or that leaf litter fall and subsequently leaf litter decomposition are not quantitatively important either, but if so, you need to argue it. This is a critical point to consider in your conceptualization.

As explained above (see R1GC2), in-stream decomposition and leaf litter in the stream are of minor importance on our experimental site. We will include this explanation in the revision.

P. 13, L. 29-31. When you talk about production and accumulation you cannot forget about mineralization. It might be that during dry and warm conditions the top soil is not hydrologically connected to the stream and thus that output of DOC from the system is non-existent, but precisely because you have those conditions you will have a larger oxygen supply and increased mineralization rates (not only increase production). This is another way in which DOC can leave the system and would need to be compared with the production term in order to estimate net accumulation. You at least need to acknowledge this.

(R1C29) We agree that during warm & dry periods, also mineralization rates should increase. Yet our observations indicate that the measured DOC in the stream during events has a high content of aromatic compounds, which are not easily mineralized. Furthermore the (top) soils of riparian zones are rich in organic matter and DOC concentration in our stream systematically increased with stream discharge during all events (see Figure 3, coefficient *a*). Both indicate that organic matter stocks are mobilization limited and provide sufficient DOC to maintain export to the stream throughout the year (Zarnetske et al., 2018). Generally we see patterns which speak for accumulation of DOC during warm & dry periods meaning that the net production is greater than the net removal rate under these circumstances. We acknowledge the referees concern and will clarify in the MS that we speak of net production.

P. 14, L. 2. Higher SUVA$_{254}$ values are commonly associated with higher aromaticity of the organic matter, rather than "processed", which might or might not be the case.
+

P. 14, L. 2-3. High SUVA$_{254}$ values representing high aromaticity together with high S$_{275-295}$ representing low molecular weight seems a bit conflicting.

(R1C30) We agree and understand the conflict. However we speak of a "relative increase in low molecular weight components" and refer to the addition of a "distinct processed riparian DOC source" as explanation for it. Hence this shall indicate that the DOC quality of deeper groundwater is very different to the riparian zone DOC quality.

For clarification we will change the MS to: "Respective, DOC quality during events changed markedly towards higher SUVA$_{254}$ values typical for higher aromaticity of the organic matter and associated to processed DOC (Hansen et al., 2016; Helms et al., 2008) and higher S$_{275-295}$ (but not as high as in cold & wet) indicating a *relative* increase in low molecular weight components in comparison to the low flow signal."

P. 14, L. 6. There are better cites than Seibert et al. (2009) for the transmissivity feedback mechanism, e.g. Bishop et al. (2004) or, originally, Rodhe (1989).

+

P. 14, L. 6-10. Which would be these preferential flow paths? Lateral water transfer through unsaturated layers over the groundwater table? Do you expect to have this process in your catchment? Do you have any groundwater table measurements in the catchment that you can plot against stream discharge to understand this?

(R1C31) We will change the citation in the MS to Bishop et al. (2004) and Rhode (1989).

Preferential flow paths may be rivulets that we observed during wet periods. They can also consist of continuous conductive structures in the subsurface that were formed by erosion and sedimentation processes caused by the meandering stream bed over the centuries. We suspect that these conduits are active when they are saturated. This leads to episodes during which DOC source areas are well connected to the streams separated by periods of poor connectivity. Direct observation of such pathways is not possible without considerable disturbance to the subsurface, which is not permitted at the site. The mild slopes in the riparian zone and the lateral distances towards the stream make it unlikely that unsaturated flow processes could deliver DOC to the stream fast enough to be consistent with the data. The depicted graph is from a groundwater well close to the study site showing that with an increase groundwater level, discharge increases in a strongly nonlinear way. This strongly hints to effective near-surface lateral drainage feeding the discharge. Unfortunately, data is only available for the last 6 months of the study period (starting at 01 July 2014), which prompted us to exclude the data in the present study. However there is a follow up study with several years of groundwater measurements at the study site. The graph will be added to the SI of the paper.

[Figure]

P. 14, L. 17. Odd grammar, please rephrase.
+
P. 14, L. 18-20. You probably have less production but you likely have less mineralization
as well which need to be accounted when discussing net accumulation.
(R1C32) Will be changed in the MS

P. 14, L. 31-32. Please, rephrase this sentence, there seems to be a verb missing or
the order of some words should be different.
(R1C33) Will be changed in the MS

P. 15, L. 5-6. But, in general, you have a positive relationship between DOC concentrations
and stream discharge and that would not support limited availability of DOC in the riparian zone.
(R1C34 ) We agree, there is a positive relationship during events and not a source limitation.
However, we want to express that lower $C_{DOC}$ values are also due to less DOC per unit water in the
riparian zone. The MS will be changed accordingly to: "Generally low $C_{DOC}$ values indicate that less
DOC is available in the riparian zone in comparison to the warm&dry situations."
P. 15, L. 7-8. Impairs both production and mineralization.
The reviewer is right and we changed this to "Unfavorable conditions for the net production of DOC
during non-event periods exist..."        ++
P. 15, L. 9-10. Exactly, because of this hydrological connectivity with rich DOC sources
I would not expect low DOC concentrations.

(R1C34) Yet there have been measured low DOC concentrations.  Yes, there is hydrological
conductivity to the DOC sources and no limitation in the source but a generally lower concentration
level as indicated in R1C34 above (P.15, L.5-6).

P. 15, L. 13. This has not been shown.
(R1C35) We agree. The MS will be changed accordingly.

P. 15, L. 14. Do your soils freeze?
(R1C35) Yes they do. However there were no clear signs of soil freezing presented in this study.
Therefore we removed this section.

P. 15, L. 15. I would argue that depletion of exportable DOC sources due to low production
is a bit too speculative as there is no information on soil and soluble pools in
the riparian zone. And again, mineralization would be low as well.
(R1C35) We agree. Depletion of exportable DOC happens because of low production in combination
with high exports as a consequence of a good hydrological connectivity (see R1C31). We will address
both, low production and hydrological connectivity in the MS.

P. 15, L. 24. Yes, I agree, the variance is low but that does not mean the absolute
values are low.
(R1C36) The absolute values are only low for $C_{DOC}$ (but not the quality parameters), presumably due
to the high amount of water in the riparian zone (leading to increased export due to hydrological
connectivity) in combination with low temperatures (leading to low production). This will be added in
the MS to: "In general, the low concentration level and low variance of DOC and the low variance of
DOC quality during winter indicates a steady export of most available source zones under relatively
low net production rates."

P. 15, L. 28-29. The lack of whether data when? Was the period prior to the beginning of the sensor measurements not recorded for weather variables and so you could not use AI$_{60}$ in your analyses after two months of sensor deployment? This needs to be clarify.

(R1C37) We agree. The MS will be changed accordingly to: "But due to the lack of weather data (the weather station was deployed two months after the sensor deployment which inhibited derivation of AI and DNT for this period), no further statements can be made for this period (Fig. 2)."

Conclusions

P. 16, L. 6-7. Again, mineralization is ignored here.

(R1C38) We agree, please refer to the comments above – we will address net production in the revision.

P. 16, L. 9-11. Exactly, wet situations are not mobilization limited and so they can lead to high DOC concentrations. And so I do not fully agree with the statement that high hydrological connectivity translate into low C$_{DOC}$, because if the source is large and you seem to have a large riparian zone, this would not be the case.

(R1C39) We agree. We meant that high hydrological connectivity leads to low C$_{DOC}$ only if the DOC net production is low compared to the DOC export but not source limited. Chances to have this situation are highest during the cold and wet situation. This will be addressed in the MS.

P. 16, L. 17. This is the only place were decomposition is acknowledged as a process potentially occurring. This needs to be taking into consideration throughout.

We agree. This will be considered in the MS.

P. 16, L. 23-27. Yes, and also actual measurements of riparian groundwater tables, riparian carbon pools and riparian soil water chemistry would be needed and helpful to understand this.

(R1C40) We agree. This will be addressed in the MS.

P. 16, L. 28-30. This sentence seems out of place.

We agree. The sentence will be moved in the MS.

P. 16, L. 31. "headwater" instead of "head water".

We agree. This will be changed in the MS.

Figures and tables

Figure 2. Maybe you can leave the dates of the x-axis only in the lower panel and remove them from the other panels (as in Figure 3). Also, a different format of the dates (e.g. MM-YY) would allow for better visualization and more data points to be characterized. Specifically the beginning and end points of the axis should be labelled.

We agree. This will be changed in the MS.

Figure 4. See previous comment on Figure 2 about the dates in the x-axis. Also, maybe thinner lines would improve visualization of the graphs.

We partly agree. X-axis will be plotted like in Figure 2. Thinner lines unfortunately did not improve visualization of the graphs.

Figure 6. My main problem with this figure is that in the warm & dry state you plot a higher $C_{DOC}$ in the soil but, again, what about the potentially high mineralization during this time. I would expect the highest $C_{DOC}$ concentrations at the end of the summer or at early autumn and specifically following warm and wet, not dry, periods.

Please consider the comments above (see R1C29, R1GC1) about net production. The warm & dry state refers to a long term hydroclimatic condition rather than an event or non-event state (see Figure 5). We will change the wording in Figure 6 to net production instead of accumulation.

  Table 1. "statistical models" instead of "models". Also, it would be helpful to know what period those descriptive statistics are based on.

This will be changed accordingly in the MS.

Table 2. All correlations are highly significant because of the large sample size. Worth mention it.

This will be changed in the MS.

Table 3. "hydroclimatic variables" instead of "environmental variables".

This will be changed in the MS.

References cited in response to Reviewer #1

Boissier, J. and Fontvieille, D.: Biodegradable dissolved organic carbon in seepage waters from two forest soils, Soil Biology and Biochemistry, 25, 1257-1261, 1993.

Boyer, J. and Groffman, P.: Bioavailability of water extractable organic carbon fractions in forest and agricultural soil profiles, Soil Biology and Biochemistry, 28, 783-790, 1996.

Creed, I. F., McKnight, D. M., Pellerin, B., Green, M. B., Bergamaschi, B., Aiken, G. R., Burns, D. A., Findlay, S. E. G., Shanley, J. B., Striegl, R. G., Aulenbach, B. T., Clow, D. W., Laudon, H., McGlynn, B. L., McGuire, K. J., Smith, R. A., and Stackpoole, S. M.: The river as a chemostat: fresh perspectives on dissolved organic matter flowing down the river continuum, Canadian Journal of Fisheries and Aquatic Sciences, 72, 1272-1285, doi:10.1139/cjfas-2014-0400, 2015.

Dawson, J. J. C., Bakewell, C., and Billett, M. F.: Is in-stream processing an important control on spatial changes in carbon fluxes in headwater catchments?, Science of The Total Environment, 265, 153-167, doi:https://doi.org/10.1016/S0048-9697(00)00656-2, 2001.

Dowell, W. H.: Kinetics and mechanisms of dissolved organic carbon retention in a headwater stream, Biogeochemistry, 1, 329-352, 1985.

Grøn, C., Tørsløv, J., Albrechtsen, H.-J., and Jensen, H. M.: Biodegradability of dissolved organic carbon in groundwater from an unconfined aquifer, Science of the Total Environment, 117, 241-251, 1992.

Huguet, A., Vacher, L., Relexans, S., Saubusse, S., Froidefond, J. M., and Parlanti, E.: Properties of fluorescent dissolved organic matter in the Gironde Estuary, Organic Geochemistry, 40, 706-719, doi:https://doi.org/10.1016/j.orggeochem.2009.03.002, 2009.

Kalbitz, K., Solinger, S., Park, J.-H., Michalzik, B., and Matzner, E.: Controls on the dynamics of dissolved organic matter in soils: a review, Soil science, 165, 277-304, 2000.

Kaplan, L. A., Wiegner, T. N., Newbold, J. D., Ostrom, P. H., and Gandhi, H.: Untangling the complex issue of dissolved organic carbon uptake: a stable isotope approach, Freshwater Biology, 53, 855-864, doi:10.1111/j.1365-2427.2007.01941.x, 2008.

Köhler, S., Buffam, I., Jonsson, A., and Bishop, K.: Photochemical and microbial processing of stream and soil water dissolved organic matter in a boreal forested catchment in northern Sweden, Aquatic Sciences, 64, 269-281, doi:10.1007/s00027-002-8071-z, 2002.

Lundquist, E. J., Jackson, L. E., and Scow, K. M.: Wet–dry cycles affect dissolved organic carbon in two California agricultural soils, Soil Biology and Biochemistry, 31, 1031-1038, doi:https://doi.org/10.1016/S0038-0717(99)00017-6, 1999.

Nelson, P., Dictor, M., and Soulas, G.: Availability of organic carbon in soluble and particle-size fractions from a soil profile, Soil Biology and Biochemistry, 26, 1549-1555, 1994.

Nimick, D. A., Gammons, C. H., and Parker, S. R.: Diel biogeochemical processes and their effect on the aqueous chemistry of streams: A review, Chemical Geology, 283, 3-17, doi:https://doi.org/10.1016/j.chemgeo.2010.08.017, 2011.

Yano, Y., McDowell, W. H., and Kinner, N. E.: Quantification of biodegradable dissolved organic carbon in soil solution with flow-through bioreactors, Soil Science Society of America Journal, 62, 1556-1564, 1998.

---

## Author Comment (AC2) · 1 Aug 2019

Dear Referee #2,

We are grateful for the thorough review of the manuscript. We will gladly make most of the changes suggested. Our new findings will be carefully classified and compared to the scientific context which will further highlight the benefits of high frequency measurements. Our response to the comment is available by the link below, with the reviewer's comment (black) quoted first and followed by our response (red).

On behalf of all co-authors,

[Figure]

Benedikt Werner

Please also note the supplement to this comment:
https://www.biogeosciences-discuss.net/bg-2019-188/bg-2019-188-AC2-supplement.pdf

———————————————————

[Figure]

**Supplement:**

Response to **Anonymous Referee #2**

"High frequency" measurements of DOC release from headwater catchments have been carried out in a number of studies before which showed broadly the same results and conclusions. For example, Broder and Biester, 2015 (also BGC) and Birkel et al 2017 published a study of high frequency measurements of DOC release from a peatland and forest catchment in the Harz (just a few kilometers away) and also modeled DOC release dependent on antecedent moisture conditions. Unfortunately, these papers have not been cited or discussed in the manuscript. What is new in this study is the really high frequency DOC monitoring (15 min) and the different statistically approaches. However, for me it is not really clear what actually the aim of the paper is. One reason for this might be that the paper lacks a clear hypothesis. Even the title does not contain a research question, just a statement of what has been done. The description of the aim of the study (p3) is quite general: : :. to obtain a better understanding: : :Looking at the conclusions, I don't think that the paper really provides more understanding than what is already known. It seems, that the authors cannot really decide if this is a eco-hydrochemical or a statistical-hydrological study. The value of the presented findings is difficult to evaluate as the authors have largely missed to compare and discuss their results to/with those of other studies. From what the authors stated in their (long) conclusions: : :.."Yet, it remains unclear which explicit mechanisms in the riparian zone are responsible for the measured and conceptualized DOC dynamics in the Rappbode stream. : : :... Further research is necessary to identify the explicit spatio-temporal mobilization patterns as well as molecular markers that can be used to trace DOC from riparian source zones into the stream in order to fully understand DOC mobilization in the riparian zone."I think that is where other studies have ended before. The biogeochemical findings in this study are quite limited, so that the study has its emphasis on the statistical approach which is clearly necessary to extract a message from the large (high frequency) data set. However, as the authors base their predictors on 60 and 30 means, the meaning of the high-frequency DOC monitoring remains form e unclear. I think it would be interesting to use this data set to evaluate which frequency is at least necessary to capture the role of the predictors and the magnitude of DOC concentration/ flux changes (38 discharge events!). Moreover, there are several factors in this data set which might be interesting to evaluate regarding the sensitivity of the model towards the predictors e.g. the magnitude of DOC-flux changes during discharge events, the role of catchment size, DOC-pools etc. but are not discussed. This manuscript is in general suitable for publication in BGC. I also think that the quality of the data and the approach is good. However, I think before this manuscript can be accepted the authors should try to give their manuscript a clearer aim/hypothesis which goes beyond a gererally better understanding of what is already known. I suggest, that the authors extend their introduction by other studies (there are numerous) on this topic. From this they can probably better deviate what is already known and what the (new) aim of their study is (why needs the frequence be higher than in other studies ?). Similar, they should extend their discussion with a comparison to data from other studies and the sensitivity and potential limitations of their predictors including the characteristics of the catchment and a discussion on high frequent high frequency should be.

(R2GC1)

We appreciate the honest opinion of Referee #2. There are four general points raised by the reviewer which we want to address to in the following.

1.1- Referee #2 recommends adding a discussion section where
- we mark down similarities and differences to other studies which broadly show the same results and conclusions
- sensitivity and potential limitations of predictors including the characteristics of the catchment should be addressed.

We agree that there have been studies carried out, which point in the same direction, but we disagree with the opinion of Referee#2: "I don't think that the paper really provides more understanding than what is already known". A comparison with other studies can help to define what is new in this study and thus will be incorporated in the MS:

a) In general most of other studies related to this topic are of a lower frequency and do not consider DOC composition. As also stated from Referee#2, to our knowledge there is no other study using a combination of "really" high frequent DOC concentration and composition time series for over one year. Yet, seeing that results of the here proposed high frequency method incorporates findings of other papers which used lower frequency measurements is an important and promising finding on its own, since it strengthens our proposed method, but also findings from other (lower frequency) papers. Please see also the discussion of 1.2 in this reply.

b) The mentioned studies from Referee#2 were conducted in a peatland which is included in a forest catchment with peaty riparian zone. Especially the DOC concentration dynamics of peatland C-Q relationships differ from that of riparian DOC mobilization dynamics (dilution vs. enrichment patterns with increasing discharge). An interesting question is if the different patterns also hold for DOC quality. Such a discussion could be fruitful, because it helps to unravel whether mobilization mechanisms are really the same in two catchments (then DOC concentration and composition dynamics in these catchments should be also comparable). In terms of DOC quality, this is not the case between these two catchments, which leads us to the conclusion that our riparian zone study site adds valuable data to complement the data for peats and peaty riparian zones provided by the studies cited by the Referee (a discussion on that will be added to the MS).

c) High frequent DOC concentration and quality is dependent on seasonal antecedent hydroclimatic changes. In order to better model and understand DOC export dynamics at such a high frequency, it is crucial to also understand the changes and interaction between the antecedent conditions at a similar time scale. This has been done in our paper and we believe highlighting the importance of continuous, interacting, hydroclimatic variables for modelling high frequency DOC data is an important contribution to former high frequency DOC export analysis. The study gives the opportunity to easily compare our findings and depicted mechanisms with other catchments which use similar (high frequency DOC concentration and composition) set ups. In terms of reproducibility, it is therefore easier to conduct in comparison to a study which uses e.g. trace metal contaminations as tracer (because such contaminations do not occur everywhere) and thus represents a potentially powerful methodological tool for examination. However, with regards to biogeochemistry and its mobilization processes, a further combination with (trace) metal export /element stoichiometry (see R#3) could turn out to be quite synergetic.

The discussion will be further complemented by a section with limitations of the predictors. In general, the statistical relationships established for predictors and DOC response cannot account for situations outside of the measurement range (extreme droughts and floodings, which are out of

calibration have to be treated with care). Furthermore, validity and sensitivity of the statistical relationships with the predictors does not account for long-term changes in biogeochemical and hydroclimatical factors (pH, ionic strength, sulfate and nitrate, annual mean temperature…) which all can influence DOC export behavior on its own.

1.2- Referee #2 recommends adding a section about the meaning and benefit of the high frequency DOC monitoring in comparison to lower frequent monitoring.

We believe an assessment about the necessity and potential of "really" high frequency measurements will highlight the findings of our study and sufficiently demarcate new findings. Within one year, DOC concentration and quality dynamics fluctuate on event and seasonal scale. Our model showed that seasonal scale drivers alone (30 d and 60 d) are able to explain the same amount of variability than hydrological event-scale drivers ($\leq$ 5d). However, it is the superposition of both which provides the more complete information to explain DOC concentration and quality dynamics throughout the year. High frequency measurements can integrate both, the high frequent part but also by (different aggregation forms) the lower frequent part of DOC variability. As presented in this paper, one can determine the optimal frequency of the low and high frequent variations, all together necessary to explain most of the DOC variance with least variables.

A comparison of our high frequency study with a low frequency study from Köhler et al. (2009) concretizes the benefit from high frequency measurements: The frequency used in our model was hourly values (~17,000 values in ~ 1.5 years) whereas Köhler et al. (2009) took 470 stream water samples in 14 years based on Köhler et al. (2008). Therefore the variance which needs to be explained shifts from a focus on seasonality and interanual variations towards high frequent fluctuations on top of the seasonal shifts. Furthermore, Köhler did not analyze the factors which are responsible for the shifts between the snow covered, melting and snow free period models. We continuously modeled discharge events throughout the whole year, as it turned out that it is exactly these shifts which could be represented by interaction of seasonal and event type predictors and they are important to understand DOC mobilization dynamics in a more holistic way through several seasons. Therefore, event based variance is needed to get better ideas of the explicit source zone activation of DOC. This frequency is in the scale of minutes to several hours.

A comparison of our high frequency and low frequency parts of our model concretizes the benefit from high frequency measurements: The Figure below shows the cumulative DOC export when just using low frequency measurements ($DNT_{30} + AI_{60} + DNT_{30} \times AI_{60}$), high frequency measurements ($Q_{hf} + Q_b + Q_{hf} \times Q_b$) or both, high and low frequency measurements. NSE of DOC export was 0.998, 0.979 and 0.783 for the "high+ low frequency", high frequency and low frequency, respectively indicating that low frequency measurements alone are not able to explain DOC export as adequate as the higher frequencies and its combination. The different export behavior of low and high frequent DOC modeling gets most pronounced during events (see Figure below).

The discussion and Figure will be analogously implemented in the manuscript and SI, respectively.

[Figure]

2- Title, aim and hypothesis (introduction) of the paper are too general.

We agree to reorganize the introduction (c.f. also Referee#3, R3GC1 and #4 R4GC1). The critique that a clear hypothesis is lacking implies the valuable suggestion to add a crisply formulated hypothesis, and we will endeavor to formulate one.

3- It seems that the authors cannot really decide if this is an eco-hydrochemical or a statistical-hydrological study.

We argue that no clear separation should be made between eco-hydrochemical and a statistical-hydrological study at such high frequency. We think both approaches are important and interacting during different hydroclimatic situations throughout the year, if viewed at high frequency.

4- The data could be used for something else

Obviously, with such a dataset there are plenty different questions to analyze. Yet we think in terms of readability, it is important to not lose the focus here. This is also why we decided to keep the biogeochemical discussion section as well as sensitivity analyses brief. Note that the data set is freely available and may be used by others (see section Data availability).

References cited in response to Reviewer #2

Köhler, S. J., Buffam, I., Laudon, H., and Bishop, K. H.: Climate's control of intra-annual and interannual variability of total organic carbon concentration and flux in two contrasting boreal landscape elements, Journal of Geophysical Research, 113, doi:10.1029/2007jg000629, 2008.
Köhler, S. J., Buffam, I., Seibert, J., Bishop, K. H., and Laudon, H.: Dynamics of stream water TOC concentrations in a boreal headwater catchment: Controlling factors and implications for climate scenarios, Journal of Hydrology, 373, 44-56, doi:10.1016/j.jhydrol.2009.04.012, 2009.

---

## Author Comment (AC3) · 1 Aug 2019

Dear Referee #3,

We thank you for the insightful comments and the help to clarify our introduction. This definitely improves the focus and thus the quality of our manuscript substantially. Our response to the comment is available by the link below, with the reviewer's comment (black) quoted first and followed by our response (red).

On behalf of all co-authors,

Benedikt Werner

[Figure]

Please also note the supplement to this comment:
https://www.biogeosciences-discuss.net/bg-2019-188/bg-2019-188-AC3-
supplement.pdf

**Supplement:**

Response to **Anonymous Referee #3**

1. There is a clear goal stated at the end of the introduction, but there are not clear hypotheses until the conceptual model is presented (Figure 6). I think that putting the conceptual framework at the front of the paper (introduction or at latest the methods) would help the reader understand how the authors are viewing the system, better appreciate the findings, and better grasp why certain methods were used.

2. The paper spends quite a bit of time discussing long-term trends in DOC attributed to changes in acid deposition, land use, and climate change. This focus was something of a red herring, as the paper is strongest on a much shorter timescale, which does not speak directly to this literature. Additionally, most of the cited papers on DOC trends are older, which I think is a recognition that while many regional trends exist (for either increasing or decreasing DOC), there is not a clear pattern or signal of anthropogenic effects on DOC concentration. There is more evidence of anthropogenic effects on DOC properties (e.g. Butman et al., 2014), and this could be fruitful, but, I think the ecohydrological focus on sources and fate of DOC is most compelling. This fits in better with the conceptual model and approach of the paper. There are many other reasons to study DOC, many of which are brought up elsewhere in the introduction (Zarnetske et al., 2018), so starting the paper with this observation is less effective.

3. The discussion seemed somewhat uneven to meâ˘AˇT with the authors still defining some concepts and findings and even describing methods. I think that reorganizing the paper around a clear set of hypotheses would strengthen this already interesting piece of work.

(R3GC1)

We appreciate your evaluation of our manuscript (MS). We acknowledge that the hypotheses of our work were not clearly stated in the introduction. Thus, we will focus the introduction more on how we see the system and mechanisms of DOC export in headwater catchments. In summary, this includes

1 - the addition of clear hypotheses, based on our conceptual framework. We reason that changes in DOC concentration and quality can greatly be explained by the hydrologic situation in the system. The DOC signal in the stream is generated by the exposure of DOC sources to mobilization. The hydrological (mobilization) and biogeochemical (production and processing) dynamics are thereby generating the runoff DOC-response. See also our response to Referee #2 (R2GC1) and #4 (R4GC1), who similarly noted the lack of a clear hypothesis.

2 - More focus on short-term dynamics in general by removing parts of the long-term DOC trend section while adding a more hydro-mechanistic point of view. We will amplify the awareness of hydrological events as a first order control on DOC dynamics. This will go hand in hand with a

3 - reorganization of the discussion section in terms of carefully reviewing the text and move methodological sections to Materials and Methods. Concept explanations which can already help to clarify the specific aim of this paper will be moved to the introduction.

We agree that all these points were not addressed clear enough in our MS as correctly pointed out by the Referee#3. We hope by addressing the above mentioned changes, we will be able to sufficiently channel the focus on the actual claims of our MS.

4. Throughout the paper, I was surprised at the lack of discussion of interactions with other elements. DOC does not cycle in isolation, and stoichiometry can have a strong influence on DOC production and consumption (Helton et al., 2015). not to find greater discussion of DOC removal mechanisms, including heterotrophic respiration and abiotic removal (Raymond et al., 2016). I imagine that nitrogen and phosphorus data are available ($NO_3^-$ data, specifically should be available through the whole time period), and including and integrating them could greatly strengthen the paper. For example, how do N and P vary during the chosen seasonal periods and how might that influence temporal patterns currently attributed to changes in source and transport limitation?

(R3GC2)
We agree that there are factors which would be useful to add understanding to the actual mobilization and production/processing/mineralization mechanisms and, as correctly mentioned by the Referee strengthen the paper. But yet we have decided to keep the focus solely on in-stream DOC quantity and quality dynamics:
Since we measured DOC in the stream, we view DOC as an integrated response signal, already carrying all the information from processing and transformation up to abiotic removal in the riparian zone. Thus, we argue that hydroclimatic dynamics are a first order control of the DOC dynamics in the stream, able to explain large proportions of the DOC variability. Based on actual measurements of the DOC dynamics, this is presented in the MS by a high correlation coefficient of hydrolclimatic variables with DOC quantity and quality as well as a high $R^2$ of our statistical models. Continual $NO_3^-$ data as well as biweekly P data is available, and it would probably allow a more in depth biogeochemical discussion, but including this data would go beyond the scope of the paper and further amplify the chance of losing focus by drifting into a more biogeochemical eco-hydrological paper. Instead, we decided to sharpen the focus only on these first order hydro-dynamical mechanisms which are considered the most dominant drivers not just in our catchment. This allows us to satisfy the - in the introduction mentioned - need to facilitate work on transferable, parsimonious models. For clarification, the above mentioned mechanisms will be discussed in the MS with the here presented point of view.

---

## Author Comment (AC4) · 1 Aug 2019

Dear Referee #4,

We would like to thank you for the thoughtful review. The comments and thoughts provided were particularly helpful to consolidate this manuscript. Our response to the comment is available by the link below, with the reviewer's comment (black) quoted first and followed by our response (red).

On behalf of all co-authors,

Benedikt Werner

Please also note the supplement to this comment:
https://www.biogeosciences-discuss.net/bg-2019-188/bg-2019-188-AC4-
supplement.pdf

──────────────────────────────

**Supplement:**

Response to **Anonymous Referee #4**

General comment

I argue that the authors have over-emphasized the trend of increasing DOC flux trends in the abstract and introduction. This is an important reason to study this subject and this work could certainly be used to better understand the mechanisms driving this trend, but the paper includes neither a report on this increasing trend or evidence for a mechanism for this trend. I think that most of the parts of the introduction are there, but I suggest that the text focus more on the aspects of DOC export reported on in the paper (transport of DOC from watersheds across hydrologic regimes and antecedent conditions). One thing that is missing from the introduction is any mention of antecedent moisture conditions. I think that discussing the role of antecedent conditions, discharge-normalized temperature, and their potential role on DOC quantity and quality should be discussed before the methods section since these are a major focus of the paper and the conceptual model discussed later (figure 6).

(R4GC1)

We agree with Referee#4, we will reorganize the introduction and add a section of antecedent (moisture) conditions to it, since they are a central focus of our findings. The introduction will be more focused on how event-scale DOC quality dynamics in headwater streams are linked to DOC mobilization processes in the riparian zone and how high-frequency measurements of $C_{DOC}$ and especially spectral properties can be utilized to identify and quantify the key controls of DOC export from event to seasonal time scales. See also our response to the general comments of Referee #2 (R2GC1) and #3 (R3GC1), who similarly raised this concern before.

Specific comments:

Page 3; Lines 14-16: I highlight this sentence because I think that it does an excellent job of encapsulating this study. I suggest the authors reorganize the introduction to better emphasize concepts related to this idea.
We agree, this describes the general claim of the study. As written above, we will reorganize the introduction towards a sharper focus on these claims (see also R4GC1).

Page 4; Line 27: Were grab samples also run for spectral slope values in addition to UVA$_{254}$? I also suggest that the authors provide some additional detail about sample collection (e.g., filter size, sample handling).
(R4C1.1) No, there were no grab samples run for spectral slope values. This was written in the results section (P. 8 L. 11-12) but a description will be added in the Methods section in the manuscript (MS). Details about samples collection are provided in the S1 section (referred to at P.4 L.25).

Page 4; Line 29: Some brief information about methods for DOC analysis (e.g. acidification level) would be of use for the reader.
(R4C1.2) We agree. The requested information will be added to the MS.

Page 5; Line 6: I suggest the authors report how closely SUVA values obtained from the sensor match the grab sample values.
(R4C1.3) Respective information can be found in the results section (P.8 L.7-10). We will reference to this in the MS.

Page 6; Line 21: I'm concerned that hysteresis loop size is biasing regression slopes obtained from this method. I would appreciate some support for application of this method to events with varying degrees of hysteresis.

Although hysteresis of C-Q relationship potentially could explain some deviations of our hydrological event models (Figure 3) indicate that the influence of hysteresis on the $R^2$ should be minor. Evaluating hysteresis index (HI) after Lloyd et al. (2016) against $R^2$ of events (Figure below) indicated a negative, but non-significant effect of magnitude of hysteresis (depicted as absolute value of HI) on $R^2$ (method of linear regression: DOC ~ Q, [DOC~log(Q) was used where appropriate). Overall, Pearson correlation of HI~$R^2$ of Events was $r^2 = 0.12$ ($r_{Pearson} = -0.34$, p = 0.07), supporting the application of our method without explicit consideration of hysteresis effects.
The discussion and Figure will be analogously implemented in the MS and SI, respectively.

[Figure]

Page 7; Line 27: Reporting an actual mean AI or similar number would be helpful for the reader.
(R4C3) We agree. The median AI60 will be added in the MS.

Page 8; Line 28: I think that there are better ways to present this information. As written, it is hard for the reader to tell what the readers should take away from this paragraph.
(R4C4) We agree. This paragraph will be clarified in the MS.

Page 9; Line 16: Presenting this information here is repeatitive. I argue this authors should move some of this material to the methods section where similar methods are already covered.
(R4C5.1, R4C5.2) We agree. Repeated information will be moved to the methods section.

Page 9; Line 30: I am concerned about the interpretation of individual regression coefficients from a multiple regression of observational data due to issues of multicollinearity. In other parts of the analysis, partial least squares regression is used to address this issue, but for this analysis it appears that multiple linear regression was used instead (Page 6, Line 18).

(R4C6) This is true. We used multiple regression analysis. However, predictors (variables and interaction terms) were tested for multicollinearity (by looking at the variance inflation factor, c.f. P.6 L.29 – P.7 L.4) and excluded from the models if there was severe multicollinearity between the predictors. This will be remarked earlier in the method section of the MS.

Page 10; Lines 9-20 and Table 3: Values of a change dramatically depending on whether a 15 or 30 day lag is used. For example, a ($C_{DOC}$) is negatively correlated to $T_{15}$, but positively correlated to T30. The same is true for $Q_{15}$ vs $Q_{30}$. This pattern is reportead for a ($SUVA_{254}$) and a ($S_{275-295}$). It would be interesting to see if the correlation between a and $DNT_{15}$ is negative. This would seem to change the implications of the study substantially.

We agree. Unfortunately when checking the correlation table again, it turned out that there was an error in the script for $T_{15}$ and $Q_{30}$, correlating the model parameter to some other/wrong variables instead. True correlation of 15 and 30 day aggregations fit together as it would be expected, hence no substantial implications and changes have to be expected from the (more in line) new correlation table. We apologize for this mistake and thank the Referee#4 for his/her thoughtful review. The mistake will be changed in the MS by replacing Table 3 by

| Model Parameters | $T_{15}$ | $T_{30}$ | $Q_{15}$ | $Q_{30}$ | $AI_6$ | $AI_{14}$ | $AI_{60}$ | $DNT_{30}$ | $Q_{hf}$ | $Q_b$ |
|---|---|---|---|---|---|---|---|---|---|---|
| z ($C_{DOC}$) | 0.05 | 0.05 | 0.02 | -0.02 | 0.05 | 0.07 | -0.09 | 0.03 | 0.15 | -0.12 |
| a ($C_{DOC}$) | 0.55 *** | 0.52 *** | -0.48 ** | -0.43 ** | -0.52 ** | **-0.65** *** | **-0.66** *** | **0.63** *** | -0.55 *** | **-0.71** *** |
| b ($C_{DOC}$) | 0.25 | 0.25 | -0.31 | -0.31 | -0.19 | -0.33 * | -0.15 | 0.32 | -0.38 * | -0.25 |
| z ($SUVA_{254}$) | 0.07 | 0.06 | 0.04 | -0.06 | -0.10 | 0.04 | -0.10 | 0.04 | 0.01 | -0.09 |
| a ($SUVA_{254}$) | 0.50 ** | 0.51 ** | -0.50 ** | -0.40 * | -0.42 ** | -0.56 *** | **-0.64** *** | 0.58 *** | -0.54 *** | **-0.60** *** |
| b ($SUVA_{254}$) | 0.21 | 0.18 | -0.32 | -0.22 | -0.10 | -0.34 * | -0.14 | 0.25 | -0.29 | -0.23 |
| z ($S_{275-295}$) | 0.00 | -0.02 | 0.21 | 0.11 | -0.09 | 0.23 | 0.04 | -0.10 | -0.02 | 0.07 |
| a ($S_{275-295}$) | **0.62** *** | **0.63** *** | -0.54 *** | -0.41 * | -0.28 | -0.47 ** | -0.56 *** | **0.62** *** | -0.47 ** | **-0.64** *** |
| b ($S_{275-295}$) | 0.13 | 0.11 | -0.31 | -0.18 | -0.12 | -0.45 ** | -0.14 | 0.19 | -0.20 | -0.24 |

Additionally, $DNT_{15}$ is in line with $DNT_{30}$ (corr. With coefficients z ($C_{DOC}$) until b ($S_{275-295}$) (from top to bottom): 0.03, 0.69***, 0.33*, 0.03, 0.65***, 0.30). Yet $DNT_{15}$ does not add any new variance to the proposed models in the paper. When replacing $DNT_{30}$ by $DNT_{15}$, R² of $C_{DOC}$, $SUVA_{254}$ and $S_{275-295}$ for the complete models drops to 0.64, 0.59 and 0.59, respectively.

Page 10; Line 28: Referring to this analysis as seasonal-scale is somewhat confusing to me because there is no specific season analysis conducted here (rather variables that AI that typically change with season are used). I would also like to know if any data were held out for model validation or if the $R^2$ statistics in Table 4 are for the model with reference to the training dataset alone.

The expression "seasonal-scale" describes the time-scale in which parameters change. This is in the time-scale of seasons (roughly 3 months) and not to be mixed up with spring, summer, fall and winter. When we speak of seasonal-scale analysis, we argue that according variables describe our hydroclimatic "seasons" in terms of "warm & dry", "cold & wet" and "intermediate". We will clarify our definition of seasonal-scale in the MS.

The $R^2$ in Table 4 refers to the complete data set models (as depicted in the referred Equation 3) for modeling DOC concentration and quality. The complete data set models were five-fold cross validated to estimate the prediction error (c.f. P.7 L.5-9). A trainings data set was only used for the PLS regression in order to derive DOC concentration from absorption spectra (see section 2.2.1).

Page 13; Lines 9-12: The different weather scenarios make sense, but I think that stating that there are three discrete states is a bit arbitrary and is not supported by any sort of data or analysis. I think the general framework is right and it makes sense for the authors to highlight certain scenarios. That said, I don't see evidence in the data for discrete states but for a continuum without jumps from one state to another. I recommend the authors clarify the nature of their conceptual model.

(R4C7) We agree. The three discrete system states were chosen for the conceptual model to highlight certain, typical scenarios out of a continuum (see also comment above). We will clarify this in the MS.

Figure 6: How baseflow DOC concentration changes with season was not supported with particular numbers in the results, but appears to be important for the conceptual framework. It is discussed briefly in a qualitative fashion, but the degree of seasonal differences in DOC concentrations are hard for the reader to infer.

We agree. We want to point out that baseflow levels under cold and wet conditions are usually higher than baseflow levels during the warm and dry phase (see Fig 5). Thus, during the cold and wet situation, higher layers of soil, more enriched in DOC get activated, but at the same time, there is also a tradeoff between amount of water (see also Referee#1, R1C31) and available DOC in the respective soil layers which can account for lower DOC concentrations. Particular median $C_{DOC}$ values were 4.13 mg L$^{-1}$, 3.72 mg L$^{-1}$ and 3.16 mg L$^{-1}$ for the warm & dry, intermediate and cold & wet state, respectively. Both warm & dry and intermediate state differ highly significant (Kruskal-Wallis test, $p <$ 0.001) from the cold & wet state. According to the significance of the different hydroclimatical situations, initial $C_{DOC}$ values of warm & dry and intermediate will be adjusted to a higher level than the cold and wet situation. Particular numbers will be integrated in the MS.

References cited in response to Reviewer #4

Lloyd, C. E. M., Freer, J. E., Johnes, P. J., and Collins, A. L.: Technical Note: Testing an improved index for analysing storm discharge-concentration hysteresis, Hydrology and Earth System Sciences, 20, 625-632, doi:10.5194/hess-20-625-2016, 2016.